# Towards Sample-Optimal Compressive Phase Retrieval with Sparse and Generative Priors

**Zhaoqiang Liu**
National University of Singapore
dcslizha@nus.edu.sg

**Subhroshekhar Ghosh**
National University of Singapore
subhrowork@gmail.com

**Jonathan Scarlett**
National University of Singapore
scarlett@comp.nus.edu.sg

## Abstract

Compressive phase retrieval is a popular variant of the standard compressive sensing problem in which the measurements only contain magnitude information. In this paper, motivated by recent advances in deep generative models, we provide recovery guarantees with near-optimal sample complexity for phase retrieval with generative priors. We first show that when using i.i.d. Gaussian measurements and an $L$-Lipschitz continuous generative model with bounded $k$-dimensional inputs, roughly $O(k \log L)$ samples suffice to guarantee that any signal minimizing an amplitude-based empirical loss function is close to the true signal. Attaining this sample complexity with a practical algorithm remains a difficult challenge, and finding a good initialization for gradient-based methods has been observed to pose a major bottleneck. To partially address this, we further show that roughly $O(k \log L)$ samples ensure sufficient closeness between the underlying signal and any *globally optimal* solution to an optimization problem designed for spectral initialization (though finding such a solution may still be challenging). We also adapt this result to sparse phase retrieval, and show that $O(s \log n)$ samples are sufficient for a similar guarantee when the underlying signal is $s$-sparse and $n$-dimensional, matching an information-theoretic lower bound. While these guarantees do not directly correspond to a practical algorithm, we propose a practical spectral initialization method motivated by our findings, and experimentally observe performance gains over various existing spectral initialization methods for sparse phase retrieval.

## 1 Introduction

In this paper, we consider the (real-valued) phase retrieval problem, which aims to recover a signal $\mathbf{x} \in \mathbb{R}^n$ from noisy magnitude-only measurements:

$$y_i = |\langle \mathbf{a}_i, \mathbf{x} \rangle| + \eta_i, \quad i = 1, 2, \ldots, m, \tag{1}$$

where $\mathbf{a}_i \in \mathbb{R}^n$ is the $i$-th sensing vector, and $\eta_i$ represents additive noise. This problem arises naturally in areas such as diffraction imaging, X-ray crystallography, microscopy, optics and astronomy [8], where it is often difficult or even impossible to observe the linear measurements directly, and one can only record the magnitudes or intensities (squared magnitudes).

In many real applications, to reduce the required number of measurements, it is of interest to exploit structure in the signal being estimated. In particular, for applications related to signal processing and imaging, it is well-known that the underlying signal typically admits a sparse or approximately sparse

35th Conference on Neural Information Processing Systems (NeurIPS 2021).

representation in some known basis [40]. Motivated by this, and considering the popularity of the standard compressive sensing (CS) problem [16], the sparse phase retrieval problem has attracted significant research interest.

Moreover, inspired by the successful applications of deep generative models in many fields [15], recently, a new perspective of CS has emerged, for which the sparsity assumption is replaced by a generative model assumption. That is, instead of assuming sparsity, the signal is assumed to be close to the range of a generative model [5]. In addition to the theoretical developments in [5], the authors also provide impressive numerical results showing that for some imaging applications, using generative priors can significantly reduce the required number of measurements (e.g., by a factor of 5 to 10) for recovering the signal up to a given accuracy. Follow-up works of [5] include [58, 22, 45, 67, 12, 29], just to name a few.

In this paper, we focus on providing recovery guarantees with near-optimal sample complexity bounds (e.g., optimal up to constant factors) for the compressive phase retrieval problem, considering both sparse and generative priors.

## 1.1 Related Work

In this subsection, we provide a summary of some relevant works, which can roughly be divided into (i) the general phase retrieval problem with no structural assumptions on the signal, (ii) sparse phase retrieval, and (iii) phase retrieval with generative models.

**General phase retrieval:** A wide range of approaches have been designed to solve the phase retrieval problem. Error-reduction approaches [18, 14] work well in practice, but they lack provable guarantees. Convex methods such as PhaseLift [9] and PhaseCut [62] lift the phase retrieval problem to a higher dimension, and typically suffer from high computation cost. Convex relaxations that operate in the natural domain of the signal are proposed in [19, 2]. However, these convex relaxation based methods are not empirically competitive against widely-used non-convex methods.

Following the seminal work of Netrapalli *et al.* [42], several works have studied theoretically-guaranteed non-convex optimization algorithms for phase retrieval [8, 10, 63, 72]. These algorithms start from a spectral initialization method, and then use an iterative algorithm (e.g., alternating minimization or gradient descent) to further decrease the approximation error. For the general phase retrieval problem, the optimal sample complexity of $O(n)$ can be achieved by certain non-convex methods [10, 63, 72]. Interestingly, optimal (up to a logarithmic factor) sample complexity guarantees have also been provided in the case of a random initialization [55, 11]. It is worth mentioning that some variants of the classic Wirtinger Flow (WF) algorithm [8, 10] and trust-region methods [55] consider squared measurements:

$$y_i = |\langle \mathbf{a}_i, \mathbf{x} \rangle|^2 + \boldsymbol{\epsilon}_i, \quad i = 1, 2, \ldots, m, \tag{2}$$

and minimize the intensity based empirical loss function:

$$f_I(\mathbf{w}) := \frac{1}{2} \left\| \mathbf{y} - |\mathbf{A}\mathbf{w}|^2 \right\|_2^2, \tag{3}$$

where $\mathbf{A} \in \mathbb{R}^{m \times n}$ is the sensing matrix with its $i$-th row being $\mathbf{a}_i^T$, and $|\cdot|$ is applied element-wise. Nonetheless, numerical results suggest that algorithms minimizing the amplitude based empirical risk (with the measurement model (1))

$$f(\mathbf{w}) := \| \mathbf{y} - |\mathbf{A}\mathbf{w}| \|_2^2 \tag{4}$$

are usually more efficient in computation [63, 72, 54]. Based on this, we focus on the measurement model (1) and the associated loss function (4).

**Sparse phase retrieval:** A variety of algorithms have been devised for sparse phase retrieval [44, 53, 50]. In particular, there exist various non-convex optimization based algorithms, including thresholded/projected WF [7, 54], sparse truncated amplitude flow [64], compressive phase retrieval with alternating minimization [28], and sparse phase retrieval by hard iterative pursuit [6]. All of these approaches are analyzed under the assumption of using a sensing matrix with i.i.d. Gaussian entries. In addition, similar to non-convex methods for the general phase retrieval problem, all of these approaches start with a spectral initialization step, and then use an iterative algorithm to refine

the initial vector. More specifically, in the initialization step, these approaches try to accurately estimate the support of the underlying signal.

Unlike the general phase retrieval setting in which optimal sample complexity guarantees are provided for some practical algorithms, for non-convex methods of sparse phase retrieval, the typical sample complexity requirement is $O(s^2 \log n)$, where $s$ is the sparsity level. This does not match the information theoretic lower bound [49, 57]; the bottleneck comes from the accurate estimation of the support set [64, 54, 28]. Improved guarantees are also available in the noiseless setting under more restrictive assumptions on the magnitudes of the non-zero entries [68]. In addition, for sparse phase retrieval, recovery guarantees with respect to *globally optimal solutions* of both the intensity and amplitude based loss functions have been provided in [13, 26, 24, 23, 69], achieving the optimal sample complexity $O(s \log n)$. However, designing a *computationally efficient algorithm achieving the optimal sample complexity* for $s$-sparse vectors remains an important open problem.

**Phase retrieval with generative models**: Phase retrieval using generative models has been studied in [20, 25, 27, 52, 66, 1]. Extensive numerical experiments for phase retrieval with generative models, as well as both Gaussian and coded diffraction pattern measurements, have been presented in [52]. Algorithms proposed in both works [20, 52] minimize the objective function directly over the latent variable $\mathbf{z} \in \mathbb{R}^k$ using gradient descent, where $k$ is the latent dimension. The corresponding objective function is highly non-convex, and performing gradient descent directly over $\mathbf{z}$ limits the explorable solution space, which may lead to getting stuck in local minima.

We highlight two particularly relevant works in more detail. To guarantee favorable global geometry for gradient methods, the authors of [20] assume a ReLU neural network generative model with i.i.d. zero-mean Gaussian weights and no offsets. In addition, the neural network needs to be sufficiently expansive such that $n_i \geq \Omega(n_{i-1} \log n_{i-1})$, where $n_i$ represents the number of neurons in the $i$-th layer. Under such conditions, a sample complexity $O(kd^2 \log n)$ is obtained, where $d$ is the number of layers. In another related work studying an approximate message passing algorithm, the authors of [1] maintain i.i.d. Gaussian weights but slightly relax to general activation functions (not only ReLU) and $n_i \geq \Omega(n_{i-1})$. Under these conditions, the high-dimensional regime is studied ($n, m, k \to \infty$ with $m/n$ kept fixed), and an asymptotic analysis is given (not yet established as fully rigorous). We also note that [1] focuses on the case that $\mathbf{z} \sim P_{\mathbf{z}}$ for some separable $P_{\mathbf{z}}$. Both [1] and [20] focus on the noiseless case, though [1] states that the extension to noisy phase retrieval is possible.

Recovery guarantees for non-convex optimization algorithms for phase retrieval with pre-trained ReLU neural network priors and untrained neural network priors have been provided in [25, 27], with assumptions of noiseless measurements and the existence of a good initial vector. Recovery guarantees for phase retrieval with generative models can also be found in [66], within a more general setting of using differentiable but unknown link functions. The weaker assumption of an unknown link function comes at a price; namely, it is an open problem to handle signals with representation error [47]. Moreover, the assumption of differentiability makes it is not directly applicable to the measurement model (1), and the loss function considered in [66] is different from both (3) and (4), which are widely adopted for phase retrieval.

## 1.2 Contributions

- We show that for compressive phase retrieval with i.i.d. Gaussian measurements, when the signal is close to the range of an $L$-Lipschitz continuous generative model with bounded $k$-dimensional inputs, roughly $O(k \log L)$ samples suffice for ensuring the closeness between the signal and any vector minimizing the amplitude-based empirical loss function (4).

- To address the sample complexity barrier posed by spectral initialization, we propose a relevant optimization problem, and show that roughly $O(k \log L)$ samples are sufficient to guarantee that any *globally optimal* solution of the optimization problem is close to the true signal. This suggests the plausibility of practical spectral initialization algorithms that are able to find such a global optimum, though we do not attempt to provide one that can provably do so.

- We adapt our analysis to the case of sparse phase retrieval, and show that roughly $O(s \log n)$ samples suffice for a similar recovery guarantee, where $s$ is the sparsity level and $n$ is the ambient dimension. This matches the information-theoretic lower bound up to constant factors

for broad signal-to-noise ratios, and up to a logarithmic factor in general; see for example [7, Theorem 3.2] and [34, Section 6.1]).

- Motivated by our theoretical findings for sparse phase retrieval, we propose a spectral initialization method based on sparse principal component analysis. We verify on synthetic experiments that our initialization method significantly outperforms several popular spectral initialization methods of sparse phase retrieval (in the sense of relative error). This further corroborates the optimal sample complexity guarantee in our theory.

## 1.3 Notation

We use upper and lower case boldface letters to denote matrices and vectors respectively. We write $[N] = \{1, 2, \cdots, N\}$ for a positive integer $N$. We define the $\ell_2$-ball $B_2^k(r) := \{\mathbf{z} \in \mathbb{R}^k : \|\mathbf{z}\|_2 \leq r\}$, and the unit sphere $\mathcal{S}^{n-1} := \{\mathbf{x} \in \mathbb{R}^n : \|\mathbf{x}\|_2 = 1\}$. We use $G$ to denote an $L$-Lipschitz continuous generative model from $B_2^k(r)$ to $\mathbb{R}^n$. For a set $B \subseteq B_2^k(r)$, we write $G(B) = \{G(\mathbf{z}) : \mathbf{z} \in B\}$. The sensing matrix $\mathbf{A} \in \mathbb{R}^{m \times n}$ is assumed to have i.i.d. standard normal entries, i.e., $a_{ij} \overset{i.i.d.}{\sim} \mathcal{N}(0, 1)$. For any $\mathbf{s} \in \mathbb{R}^n$, we use $\bar{\mathbf{s}} = \frac{\mathbf{s}}{\|\mathbf{s}\|_2}$ to denote the corresponding normalized vector. The support (set) of a vector is the index set of its non-zero entries. For any $\mathbf{X} \in \mathbb{R}^{m \times n}$ and any index set $I \subseteq [m]$, $\mathbf{X}_I$ denotes the $|I| \times n$ sub-matrix of $\mathbf{X}$ that only keeps the rows indexed by $I$. We use $\|\mathbf{X}\|_{2 \to 2}$ to denote the spectral norm of a matrix $\mathbf{X}$. For any positive integer $N$, we use $\mathbf{I}_N$ to denote the identity matrix in $\mathbb{R}^{N \times N}$. We use the generic notations $C$ and $C'$ to denote large positive constants, and we use $c$ and $c'$ to denote small positive constants; their values may differ from line to line.

## 2 Amplitude-Based Loss Minimization for Generative Priors

In this section, we provide recovery guarantees with respect to optimal solutions of the amplitude based loss function (4) for phase retrieval with generative models. We let the generative model $G : B_2^k(r) \to \mathbb{R}^n$ be any $L$-Lipschitz continuous function, and we suppose that the underlying signal $\mathbf{x}$ is close to (but does not need to lie in) the range of $G$. We are interested in the case $k \ll n$ (i.e., relatively small latent dimension). The sensing matrix $\mathbf{A}$ is assumed to have i.i.d. standard normal entries. Let $\mathbf{q} \in \mathrm{Range}(G)$ minimize (4) to within additive $\tau > 0$ of the optimum, i.e.,

$$\|\mathbf{y} - |\mathbf{A}\mathbf{q}|\|_2^2 \leq \min_{\mathbf{w} \in \mathrm{Range}(G)} \|\mathbf{y} - |\mathbf{A}\mathbf{w}|\|_2^2 + \tau, \tag{5}$$

and define $\mathbf{p} \in \mathrm{Range}(G)$ to be the point that is closest to the signal $\mathbf{x}$, i.e.,

$$\mathbf{p} = \arg \min_{\mathbf{w} \in \mathrm{Range}(G)} \|\mathbf{x} - \mathbf{w}\|_2. \tag{6}$$

With the above settings in place, we present the following theorem, proved in Appendix B.

**Theorem 1.** *Consider the observation model* (1) *with i.i.d. Gaussian measurements, and* $\mathbf{p}, \mathbf{q}$ *satisfying* (5)–(6) *with parameter* $\tau$. *Then, for any* $\delta > 0$, *we have that if* $m = \Omega(k \log \frac{Lr}{\delta})$,[1] *with probability* $1 - e^{-\Omega(m)}$, *it holds that*

$$\min\{\|\mathbf{q} - \mathbf{x}\|_2, \|\mathbf{q} + \mathbf{x}\|_2\} \leq O\left(\|\mathbf{p} - \mathbf{x}\|_2 + \frac{\|\boldsymbol{\eta}\|_2 + \sqrt{\tau}}{\sqrt{m}} + \delta\right). \tag{7}$$

Since we assume that $\mathbf{x}$ is close to the range of $G$, the representation error $\|\mathbf{p} - \mathbf{x}\|_2$ is small (compared to $\|\mathbf{x}\|_2$). The optimization error $\tau$ is also ideally small. In addition, it is common to assume that $\frac{\|\boldsymbol{\eta}\|_2}{\sqrt{m}} \leq c\|\mathbf{x}\|_2$ (e.g., see [72]), where $c > 0$ is also small. A typical $d$-layer neural network generative model has $\mathrm{poly}(n)$-bounded weights in each layer, and thus its Lipschitz constant is $L = n^{O(d)}$ [5]. Then, the values of $r$ and $\delta$ can be as extreme as $r = n^{O(d)}$ and $\delta = \frac{1}{n^{O(d)}}$ without affecting the final scaling. In this case, Theorem 1 reveals that roughly $O(k \log L)$ samples suffice to guarantee that any optimal solution of the amplitude-based loss function (4) is close to the true signal (up to an unavoidable possible global sign flip). By the analysis of information-theoretic lower bounds in [31, 37], this scaling cannot be improved without further assumptions.

---

[1]Here and subsequently, all $m = \Omega(\cdot)$ statements are assumed to have a large enough implied constant.

# 3   Spectral Initializations with Generative Priors

In Theorem 1, we provided recovery guarantees with respect to optimal solutions of the amplitude based loss function. However, it is not clear whether such a guarantee is achievable by practical algorithms.

A notable recent work is [25], showing that for a depth-$d$, width-$w$ ReLU neural network, roughly $O(kd \log w)$ samples[2] suffice for their alternating phase projected gradient descent (APPGD) approach to obtain an accurate estimate of the signal, as long as there exists a proper initial vector $\mathbf{x}^{(0)}$, in the sense that $\min\{\|\mathbf{x} - \mathbf{x}^{(0)}\|_2, \|\mathbf{x} + \mathbf{x}^{(0)}\|_2\} \le c\|\mathbf{x}\|_2$ for some small positive constant $c$. However, as we know from sparse phase retrieval, designing a practical algorithm to find a proper initial vector with near-optimal sample complexity can be more difficult than designing the subsequent iterative algorithm that refines the initial guess, and this still remains as an open problem. It is also important to note that [25] assumes accurate projections onto the range of the generative model, but this is not guaranteed in practice, due to the use of approximations [51, 48].

Existing initialization methods for sparse phase retrieval typically first estimate the support of the sparse signal, and then perform a power method on a sub-matrix corresponding to the estimated support set to calculate an initial vector. Without extra assumptions beyond sparsity, these methods have a suboptimal sample complexity of $O(s^2 \log n)$. For phase retrieval with generative models, the situation is even worse – there is no theoretically-guaranteed initialization method at all. To address this gap, we provide recovery guarantees for a spectral initialization method of phase retrieval with generative models. We emphasize that our guarantees only concern the global optima of a suitably-defined optimization problem; see (14) below. Similarly to [25], the caveat remains that practical solutions may fail to find such optima (e.g., due to inexact projections).

We assume that $\mathbf{x}$ is in the range of $G : B_2^k(r) \to \mathbb{R}^n$ (again assumed to be $L$-Lipschitz continuous), and we assume that the noise terms $\eta_1, \eta_2, \ldots, \eta_m$ are bounded as follows:

$$\frac{\|\boldsymbol{\eta}\|_2}{\sqrt{m}} \le c_0\|\mathbf{x}\|_2, \tag{8}$$

where $c_0 > 0$ is a sufficiently small constant. This assumption states that the signal to noise ratio (SNR) is sufficiently large, and has also been made in relevant existing works (see, e.g., [72, Theorem 3]). Moreover, we assume that

$$\|\boldsymbol{\eta}\|_\infty \le c_1\|\mathbf{x}\|_2, \tag{9}$$

where $c_1 > 0$ is also a sufficiently small constant. This assumption is satisfied, for example, when (assuming (8)),

$$\|\boldsymbol{\eta}\|_\infty = O\left(\frac{\|\boldsymbol{\eta}\|_2}{\sqrt{m}}\right), \tag{10}$$

i.e., when none of the noise entries are unusually large. Again, similar assumptions have been made in existing works such as [10, Theorem 2], and [71, Theorem 2].[3]

Let $\lambda$ be defined as

$$\lambda := \sqrt{\frac{\pi}{2}} \cdot \frac{1}{m} \sum_{i=1}^m y_i. \tag{11}$$

Using sub-Gaussian concentration [60, Proposition 5.10], $\lambda$ is close to $\|\mathbf{x}\|_2$ with high probability given enough samples. In the following, we focus on estimating the normalized signal vector $\bar{\mathbf{x}} := \frac{\mathbf{x}}{\|\mathbf{x}\|_2}$. For this purpose, similar to the idea in [35], we consider a normalized generative model $\tilde{G} : \mathcal{D} \to \mathcal{S}^{n-1}$, where $\mathcal{D} := \{\mathbf{z} \in B_2^k(r) : \|G(\mathbf{z})\|_2 > R_{\min}\}$ for some $R_{\min} > 0$,[4] $\mathcal{S}^{n-1}$ denotes the unit sphere in $\mathbb{R}^n$, and $\tilde{G}(\mathbf{z}) = \frac{G(\mathbf{z})}{\|G(\mathbf{z})\|_2}$. Then, $\tilde{G}$ is $\tilde{L}$-Lipschitz continuous with $\tilde{L} = \frac{L}{R_{\min}}$.

---

[2]This matches the $O(k \log L)$ scaling upon substituting $L = O(w^d)$ for ReLU networks [5].

[3]When $\boldsymbol{\eta}$ is Gaussian with $\boldsymbol{\eta} \sim \mathcal{N}(\mathbf{0}, \sigma^2 \mathbf{I}_m)$, (10) is not guaranteed, but any "truncated Gaussian" does satisfy (10). In addition, to circumvent this issue under Gaussian noise, we may instead assume that $\sigma\sqrt{\log m} \le c_0\|\mathbf{x}\|_2$ in (8) (and remove the assumption in (9)). We also note that in [10, 71], the authors consider an intensity based measurement model, and accordingly, the corresponding assumption is for $\frac{\|\boldsymbol{\eta}\|_\infty}{\|\mathbf{x}\|_2^2}$ instead of for $\frac{\|\boldsymbol{\eta}\|_\infty}{\|\mathbf{x}\|_2}$.

[4]As discussed in [35], the dependence on $R_{\min}$ in the sample complexity is very mild. Under the typical scaling $L = n^{O(d)}$ for a $d$-layer neural network, the scaling laws remain unchanged even with $R_{\min} = \frac{1}{n^{O(d)}}$.

In the following, we adopt an idea from [72] of considering an empirical matrix with a truncation operation. We note that [72] only considered doing so for general phase retrieval; we are not aware of any work doing so for constrained settings (e.g., with sparse or generative priors), and we found that handling such settings required substantial additional effort. In more detail, we consider the following matrix defined with a truncation operation (see Remark 2 below for a discussion comparing to the non-truncated counterpart):

$$\mathbf{V} := \frac{1}{m} \sum_{i=1}^{m} y_i \mathbf{a}_i \mathbf{a}_i^T \mathbf{1}_{\{l\lambda < y_i < u\lambda\}}, \tag{12}$$

where $l, u$ are positive constants satisfying $1 < l < u$, and $\lambda$ is given in (11). Note that for the noiseless case with $y_i = |\langle \mathbf{a}_i, \mathbf{x} \rangle|$, in accordance with the above-mentioned closeness between $\lambda$ and $\|\mathbf{x}\|_2$, it is useful to consider the following expectation:

$$\mathbf{J} := \mathbb{E}\left[ \frac{1}{m} \sum_{i=1}^{m} y_i \mathbf{a}_i \mathbf{a}_i^T \mathbf{1}_{\{l\|\mathbf{x}\|_2 < y_i < u\|\mathbf{x}\|_2\}} \right] = \|\mathbf{x}\|_2 (\gamma_0 \mathbf{I}_n + \beta_0 \bar{\mathbf{x}} \bar{\mathbf{x}}^T), \tag{13}$$

where for $g \sim \mathcal{N}(0, 1)$, we define $\gamma_0 := \mathbb{E}[|g| \mathbf{1}_{\{l < |g| < u\}}]$ and $\beta_0 := \mathbb{E}[|g|^3 \mathbf{1}_{\{l < |g| < u\}}] - \gamma_0$ (*cf.* Lemma 8 in Appendix C.2). Based on the observation that $\bar{\mathbf{x}}$ is a leading eigenvector of $\mathbf{J}$ and the assumption $\mathbf{x} \in \text{Range}(G)$, we consider using the following optimization problem to find an $\hat{\mathbf{x}}$ that approximates $\bar{\mathbf{x}}$:

$$\hat{\mathbf{x}} := \arg \max_{\mathbf{w} \in \tilde{G}(\mathcal{D})} \mathbf{w}^T \mathbf{V} \mathbf{w}. \tag{14}$$

In addition, in order to take the norm of $\mathbf{x}$ into account, we set the the initial vector $\mathbf{x}^{(0)}$ as

$$\mathbf{x}^{(0)} = \lambda \hat{\mathbf{x}}. \tag{15}$$

We show that roughly $O(k \log L)$ samples suffice for ensuring the closeness between $\mathbf{x}^{(0)}$ and $\mathbf{x}$ (up to a global signal flip). Formally, we have the following.

**Theorem 2.** *Consider the model* (1) *with i.i.d. Gaussian measurements and noise satisfying* (8)–(9), *and let $\hat{\mathbf{x}}$ be as defined in* (14). *Suppose that[5] $\bar{\mathbf{x}} \in \tilde{G}(\mathcal{D})$ and $u > l > 1 + c_1$, where $c_1 \geq \frac{\|\boldsymbol{\eta}\|_\infty}{\|\mathbf{x}\|_2}$ appears in* (9). *Given a sufficiently small positive constant $c$,[6] when $m = \Omega(k \log(\tilde{L}nr))$, we have with probability $1 - e^{-\Omega(m)}$ that*

$$\min\{\|\hat{\mathbf{x}} - \bar{\mathbf{x}}\|_2, \|\hat{\mathbf{x}} + \bar{\mathbf{x}}\|_2\} < 0.9c. \tag{16}$$

*In addition, we have with probability $1 - e^{-\Omega(m)}$ that $\mathbf{x}^{(0)} = \lambda \hat{\mathbf{x}}$ satisfies*

$$\min\{\|\mathbf{x}^{(0)} - \mathbf{x}\|_2, \|\mathbf{x}^{(0)} + \mathbf{x}\|_2\} < c\|\mathbf{x}\|_2. \tag{17}$$

The proof is given in Appendix C. Using well-established chaining arguments as those in [5, 35, 36], the sample complexity $\Omega(k \log(\tilde{L}nr))$ can be reduced to $\Omega(k \log(\tilde{L}r))$. However, since for a typical $d$-layer neural network, we have the Lipschitz constant $L = n^{O(d)}$ (and thus $\tilde{L}$ and $r$ can also be of order $n^{O(d)}$) [5], such a reduction is typically of minor importance.

## 4 Sparse Phase Retrieval

The proof of Theorem 2 relies on the fact that $\bar{\mathbf{x}}$ lies in $\text{Range}(\tilde{G}) \subseteq \mathcal{S}^{n-1}$, with the $\delta$-covering number being upper bounded by $\left(\frac{\tilde{L}r}{\delta}\right)^k$. Letting $\Sigma_s^n$ be the set of all $s$-sparse unit vectors in $\mathbb{R}^n$, we know that the $\delta$-covering number of $\Sigma_s^n$ is upper bounded by $\binom{n}{s}\left(\frac{1}{\delta}\right)^s \leq \left(\frac{en}{\delta s}\right)^s$ [3]. Based on this observation, we can readily adapt the proof of Theorem 2 to obtain the following theorem.

---

[5]This is weaker than the assumption that $\|\mathbf{x}\|_2 > R_{\min}$ made in [35]. Moreover, even if $\|\mathbf{x}\|_2 \leq R_{\min}$, the condition $\bar{\mathbf{x}} \in \tilde{G}(\mathcal{D})$ may be satisfied (e.g., when $G$ is a ReLU neural network generative model with no offsets, then the range of $G$ becomes a cone, and if $0 < \|\mathbf{x}\|_2 \leq R_{\min}$, we can choose a sufficiently large $C$ such that $C\|\mathbf{x}\|_2 > R_{\min}$, with $C\mathbf{x} \in \text{Range}(G)$).

[6]Note that the smaller $c$ is taken to be here, the smaller the constant $c_0$ needs to be in the assumption (8).

**Theorem 3.** *Consider the model* (1) *with i.i.d. Gaussian measurements and noise satisfying* (8)–(9), *and suppose that* $\mathbf{x} \in \mathbb{R}^n$ *is $s$-sparse. Let $\lambda$ be defined as in* (11). *Suppose that $u > l > 1 + c_1$, where $c_1 \geq \frac{\|\boldsymbol{\eta}\|_\infty}{\|\mathbf{x}\|_2}$ appears in* (9), *and $\mathbf{V}$ is defined in* (12). *Let $\hat{\mathbf{x}}$ be defined as*

$$\hat{\mathbf{x}} := \arg \max_{\mathbf{w} \in \Sigma_s^n} \mathbf{w}^T \mathbf{V} \mathbf{w}. \tag{18}$$

*Then, when $m = \Omega\big(s \log n\big)$, we have with probability $1 - e^{-\Omega(m)}$ that*

$$\min\{\|\hat{\mathbf{x}} - \bar{\mathbf{x}}\|_2, \|\hat{\mathbf{x}} + \bar{\mathbf{x}}\|_2\} < 0.9c, \tag{19}$$

*where $c > 0$ is a sufficiently small constant. In addition, we have with probability $1 - e^{-\Omega(m)}$ that $\mathbf{x}^{(0)} = \lambda \hat{\mathbf{x}}$ satisfies*

$$\min\{\|\mathbf{x}^{(0)} - \mathbf{x}\|_2, \|\mathbf{x}^{(0)} + \mathbf{x}\|_2\} < c\|\mathbf{x}\|_2. \tag{20}$$

**Implications regarding SPCA and support recovery:** We note that (18) is essentially a problem of sparse principal component analysis (SPCA) [73, 38]. Existing spectral initialization methods for sparse phase retrieval typically start from estimating the support of $\mathbf{x}$, with an accurate estimate of the support requiring a suboptimal $O(s^2 \log n)$ sample complexity [7, 64, 28]. This serves as the major bottleneck of the sample complexity upper bound of the whole procedure (i.e., the initialization step and the subsequent iterative step), and it is unknown whether there is a practical and sample-optimal approach for estimating the support. In fact, to our knowledge, even the following question has not been answered previously:

Key question: *Does there exist any spectral-based method (computationally efficient or otherwise) attaining an accurate initial vector with near-optimal sample complexity?*

We provide a positive answer to this question, showing that $O(s \log n)$ samples suffice to ensure that any optimal solution of the SPCA problem (18) is close to the underlying (normalized) signal.

Motivated by the strong sample complexity guarantee provided in Theorem 3, we propose a spectral initialization method for sparse phase retrieval as described in Algorithm 1. In particular, after calculating $\mathbf{V}$ from $\mathbf{A}$ and $\mathbf{y}$, we use algorithms for SPCA to solve (18). We observe via numerical experiments in Section 5 that our initialization method outperforms several popular spectral initialization methods used for sparse phase retrieval.

**Remark 1.** *In various works on SPCA, it has been shown that there exists a statistical-computational gap for achieving consistency and optimal convergence rates uniformly over a parameter space ( e.g., see [4, 65, 7]). However, the models therein are all different from ours (e.g., the spiked covariance model [17]), so it is unclear to what extent we should expect similar computational difficulties.*

**Remark 2.** *The truncation function $\mathbf{1}_{\{l\lambda < y_i < u\lambda\}}$ in* (12) *plays an important role in our analysis. Specifically, it allows us to apply sub-exponential concentration bounds to certain quantities of interest, where the non-truncated version would involve heavier-tailed random variables (e.g., sub-Weibull of order $\alpha < 1$ [21, Theorem 3.1]). We believe that this distinction could be even more important when seeking uniform recovery guarantees, though we do not do so in this work. In Section 5, we present numerical results for applying SPCA to the non-truncated weighted empirical matrix*

$$\tilde{\mathbf{V}} = \frac{1}{m} \sum_{i=1}^m y_i \mathbf{a}_i \mathbf{a}_i^T. \tag{21}$$

*We will see in our experiments that the method corresponding to the truncated empirical matrix $\mathbf{V}$ outperforms the method corresponding to the non-truncated empirical matrix $\tilde{\mathbf{V}}$, although the expectations of these two matrices have a very similar structure. Thus, we believe that truncation may potentially help in overcoming the computational challenges discussed above.*

We note that [43, 56] consider solving sparse phase retrieval via practical SPCA methods, in a more general setting considering an unknown link function. They consider a weighted empirical covariance matrix with no truncation, and the derived sample complexity has a quadratic dependence on $s$.

**Algorithm 1** A spectral initialization method for sparse phase retrieval based on SPCA (`PRI-SPCA`)

---
**Input**: $\mathbf{A}$, $\mathbf{y}$, $l$, $u$
**Output**: An initial vector $\mathbf{x}^{(0)}$
1) Calculate $\lambda$ from (11) and then calculate $\mathbf{V}$ from (12).
2) Use an SPCA algorithm to compute $\hat{\mathbf{x}}$ according to (18).
3) Let $\mathbf{x}^{(0)} = \lambda\hat{\mathbf{x}}$.

---

# 5 Numerical Experiments

In this section, we perform synthetic experiments to complement our theoretical results. We focus only on the sparse setting, in part because numerous existing algorithms are known for sparse priors but not for generative priors. Having said this, generative priors come with their own unique practical challenges (e.g., approximate projection methods), and we believe that their empirical study and associated practical variations would also be of significant interest for future work.

We compare our method (`PRI-SPCA`; see Algorithm 1) with spectral initialization methods used in several popular algorithms for sparse phase retrieval: Thresholded Wirtinger flow (`ThWF`) [7], sparse truncated amplitude flow (`SPARTA`) [64], and compressive phase retrieval with alternating minimization (`CoPRAM`) [28].[7] We also compare with the approach of solving an SPCA problem with respect to $\tilde{\mathbf{V}}$ (*cf.* (21)), which is a non-truncated version of $\mathbf{V}$. The corresponding method is termed `PRI-SPCA-NT`, with 'NT' standing for no truncation. For `PRI-SPCA`, we set $l = 1$ and $u = 5$, as previously suggested in [72].

Various algorithms have been developed to tackle the SPCA problem, see, e.g., [41, 30, 39]. In our experiments, we present the numerical results using the generalized Rayleigh quotient iteration (`GRQI`) method [33], but we observed that other popular SPCA methods such as the truncated power method (`TPower`) [70] give similar averaged relative errors.[8] For `GRQI`, we set the deflation parameter to be $0.2$, and the total number of iterations to be $100$. Since the sparsity level $s$ is typically assumed to be known a priori for `SPARTA` and `CoPRAM` (`ThWF` assumes the knowledge of a parameter that plays the similar role as $s$), for `GRQI`, we set the parameter about the maximum number of non-zero indices as $s$. Note that the time complexity of each iteration of `GRQI` is $O(ns + s^3)$, which scales mildly compared to the subsequent $O(s^2 n \log n)$ time complexity of the iterative methods used in the three approaches (for `ThWF`, the corresponding time complexity is $O(n^2 \log n)$. See [28, Table I]). The number of power method steps used in all the methods is fixed to be $100$.

The authors of [33] suggest using the column of maximal norm as a starting point of `GRQI`. Because this policy has a time complexity of $O(n^2)$, we instead choose the column corresponding to the largest diagonal entry as the starting point. Due to the structure of $\mathbf{V}$ (*cf.* (13)), this modified strategy is very similar to the suggested policy, but with a smaller time complexity requirement. For a fair comparison, all power methods used in `ThWF`, `SPARTA`, and `CoPRAM` are similarly initialized. For the initialization method used in `ThWF`, we set the tuning parameter as $\alpha = 0.1$, as suggested in [7]. Similarly, we set $|\bar{\mathcal{I}}| = \lceil \frac{1}{6} m \rceil$ for the initialization method of `SPARTA`.

The support of the signal vector $\mathbf{x}$ is uniformly random, and the non-zero entries are generated from i.i.d. standard normal distributions. The noise vector $\boldsymbol{\eta}$ is generated from the distribution $\mathcal{N}(\mathbf{0}, \sigma^2 \|\mathbf{x}\|_2^2 \mathbf{I}_m)$, where $\sigma > 0$ represents the noise level. The initial relative error is defined as

$$\frac{\min\{\|\mathbf{x} - \mathbf{x}^{(0)}\|_2, \|\mathbf{x} + \mathbf{x}^{(0)}\|_2\}}{\|\mathbf{x}\|_2}, \tag{22}$$

where $\mathbf{x}^{(0)}$ is an initial vector produced by each of the above mentioned initialization methods. The reported relative error is averaged over 50 random trials, and error bars indicate a single standard deviation over these trials. All numerical experiments are conducted using MATLAB R2014b on a machine with an Intel Core i5 CPU at 1.8 GHz and 8GB RAM.

---

[7]We use the MATLAB package shared by the authors of [28] at `https://github.com/GauriJagatap/model-copram`. The package not only contains the codes for `CoPRAM`, but also for `ThWF` and `SPARTA`.

[8]Using `TPower` also results in small relative error for our `PRI-SPCA` method, but in general, we found it to be less stable than `GRQI` in the absence of a careful design of its starting point.

1. **Sample size effect:** In our first experiment, similarly to [64], we consider noiseless measurements, and we fix the signal dimension as $n = 1000$. We consider $s$ equaling both 10 and 20. The number of measurements $m$ takes values in $\{100, 200, \ldots, 3000\}$. From Figure 1, we observe that our `PRI-SPCA` method almost always achieves the smallest initial relative error, thus outperforming all other methods, including the `PRI-SPCA-NT` method, which uses the non-truncated empirical matrix $\tilde{\mathbf{V}}$.

2. **Sparsity effect**: We consider noiseless measurements and fix $n = 1000$, and set $m = 1000$ or 2000, and vary $s$ in $\{5, 10, \ldots, 50\}$. We observe from Figure 2 that for this experiment, our `PRI-SPCA` method also gives the best empirical performance.

3. **Noise effect**: To demonstrate that `PRI-SPCA` also performs well in the noisy case, we vary $\sigma$ in $\{0.1, 0.2, \ldots, 1\}$. In addition, similarly to [64], we fix $n = 1000$, $m = 3000$, and consider $s$ equaling both 10 and 20. For this experiment, similarly to that in [28], we do not compare with `ThWF`, as it uses quadratic measurements. From Figure 3, we observe that `PRI-SPCA` outperforms `SPARTA` and `CoPRAM`, and it is better than `PRI-SPCA-NT` at low noise levels.

In Appendix E, we additionally compare with other initialization methods in terms of the effectiveness when combining with the subsequent iterative algorithm of `CoPRAM`. In particular, we compare the relative error and empirical success rate in the noiseless case, whereas for the noisy case, we compare the relative error when using approximately the same running time. We observe that for most cases, our `PRI-SPCA` method is superior to all other methods we compare with. This again suggests that `PRI-SPCA` can serve as a good initialization method in practice.

## 6   Conclusion and Future Work

We have provided a near-optimal recovery guarantee for amplitude-based loss minimization for phase retrieval with generative priors. In addition, motivated by the bottleneck of the spectral initialization for compressive phase retrieval, we have provided near-optimal recovery guarantees with respect to the optimal solutions of optimization problems designed for spectral initialization. An immediate future research direction is to design provably sample-optimal *and computationally efficient* spectral initialization methods that build on our theoretical results. In addition, extensions to complex-valued phase retrieval [32, 46] would be of significant interest.

**Acknowledgment.** This work was supported by the Singapore National Research Foundation (NRF) under grant R-252-000-A74-281, and the Singapore Ministry of Education (MoE) under grants R-146-000-250-133 and R-146-000-312-114.

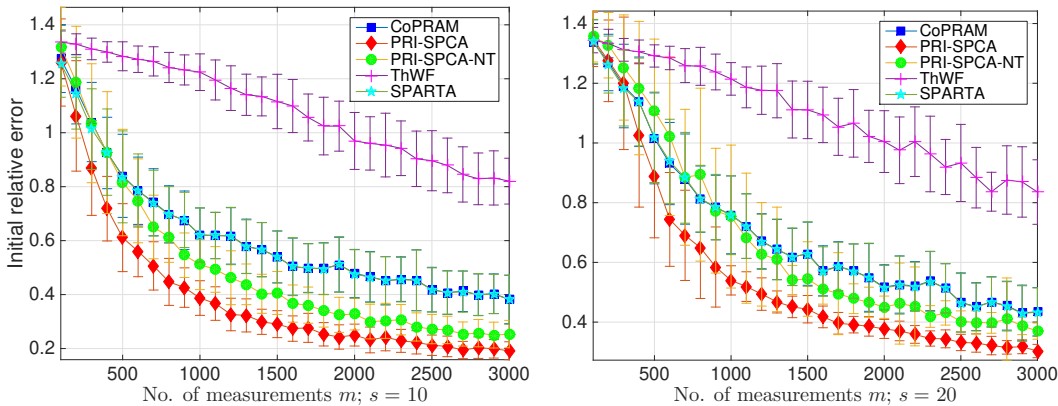

Figure 1: Average relative error vs. number of measurements $m$ with $s = 10$ (Left), $s = 20$ (Right).

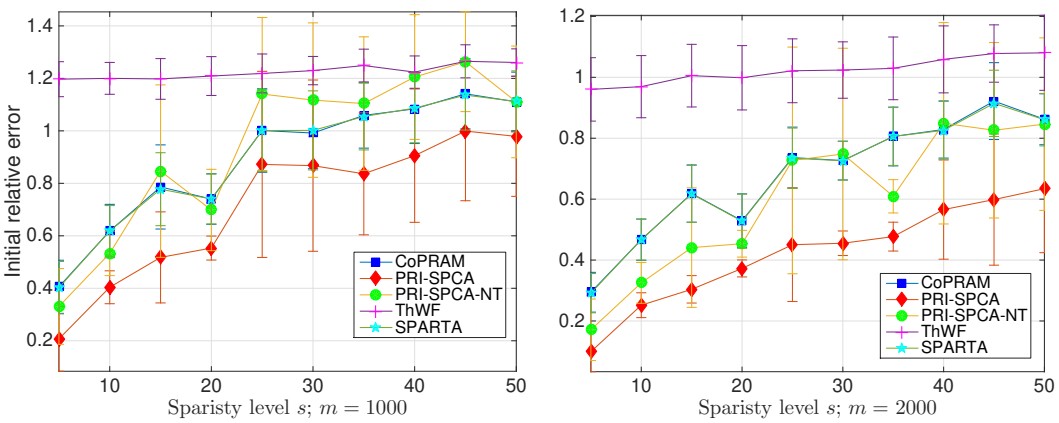

Figure 2: Average relative error vs. sparsity level $s$ with $m = 1000$ (Left), $m = 2000$ (Right).

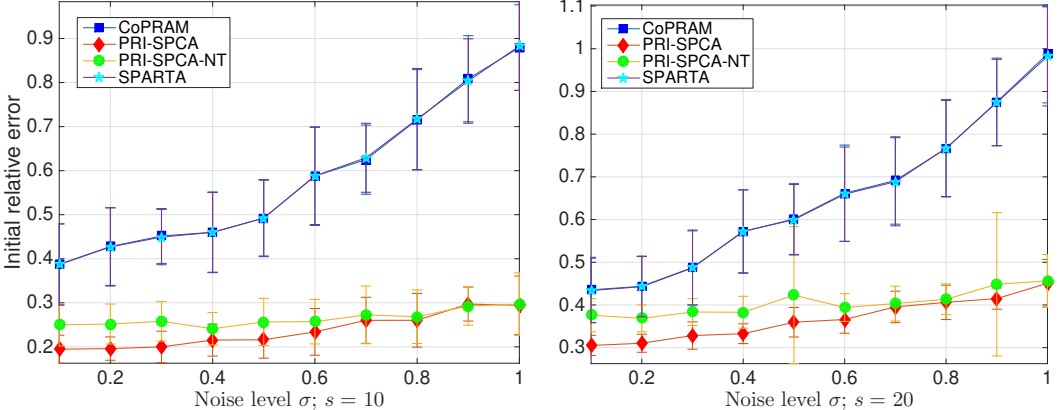

Figure 3: Average relative error vs. noise level $\sigma$ with $s = 10$ (Left), $s = 20$ (Right).

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
