# Supplementary Material

## Towards Sample-Optimal Compressive Phase Retrieval with Sparse and Generative Priors (NeurIPS 2021)

Zhaoqiang Liu, Subhroshekhar Ghosh, and Jonathan Scarlett

## A    General Auxiliary Results

In this section, we provide some useful auxiliary results that will be used throughout our analysis. First, we have the following basic concentration inequality.

**Lemma 1.** ([59, Lemma 1.3]) *Suppose that $\mathbf{A} \in \mathbb{R}^{m \times n}$ has i.i.d. standard normal entries. For fixed $\mathbf{x} \in \mathbb{R}^n$, we have for any $\epsilon \in (0, 1)$ that*

$$\mathbb{P}\left((1-\epsilon)\|\mathbf{x}\|_2^2 \le \left\|\frac{1}{\sqrt{m}}\mathbf{A}\mathbf{x}\right\|_2^2 \le (1+\epsilon)\|\mathbf{x}\|_2^2\right) \ge 1 - 2e^{-\epsilon^2(1-\epsilon)m/4}. \tag{23}$$

Next, we state the following standard definition.

**Definition 1.** *A random variable $X$ is said to be sub-Gaussian if there exists a positive constant $C$ such that $(\mathbb{E}\left[|X|^p\right])^{1/p} \le C\sqrt{p}$ for all $p \ge 1$. The sub-Gaussian norm of a sub-Gaussian random variable $X$ is defined as $\|X\|_{\psi_2} = \sup_{p \ge 1} p^{-1/2}\left(\mathbb{E}\left[|X|^p\right]\right)^{1/p}$.*

According to [60, Proposition 5.10], we have the following concentration inequality for sub-Gaussian random variables.

**Lemma 2.** (Hoeffding-type inequality [60, Proposition 5.10])  *Let $X_1, \ldots, X_N$ be independent zero-mean sub-Gaussian random variables, and let $K = \max_i \|X_i\|_{\psi_2}$.  Then, for any $\boldsymbol{\alpha} = [\alpha_1, \alpha_2, \ldots, \alpha_N]^T \in \mathbb{R}^N$ and any $t \ge 0$, it holds that*

$$\mathbb{P}\left(\left|\sum_{i=1}^N \alpha_i X_i\right| \ge t\right) \le \exp\left(1 - \frac{ct^2}{K^2\|\boldsymbol{\alpha}\|_2^2}\right), \tag{24}$$

*where $c > 0$ is a constant.*

Alongside the sub-Gaussian notion in Definition 1, we use the following definition of a sub-exponential random variable and sub-exponential norm.

**Definition 2.** *A random variable $X$ is said to be sub-exponential if there exists a positive constant $C$ such that $(\mathbb{E}\left[|X|^p\right])^{\frac{1}{p}} \le Cp$ for all $p \ge 1$. The sub-exponential norm of $X$ is defined as $\|X\|_{\psi_1} = \sup_{p \ge 1} p^{-1}\left(\mathbb{E}\left[|X|^p\right]\right)^{\frac{1}{p}}$.*

We have the following concentration inequality for sums of independent sub-exponential random variables.

**Lemma 3.** (Bernstein-type inequality [60, Proposition 5.16]) *Let $X_1, \ldots, X_N$ be independent zero-mean sub-exponential random variables, and $K = \max_i \|X_i\|_{\psi_1}$. Then for every $\boldsymbol{\alpha} = [\alpha_1, \ldots, \alpha_N]^T \in \mathbb{R}^N$ and $\epsilon \ge 0$, it holds that*

$$\mathbb{P}\left(\left|\sum_{i=1}^N \alpha_i X_i\right| \ge \epsilon\right) \le 2\exp\left(-c \cdot \min\left(\frac{\epsilon^2}{K^2\|\boldsymbol{\alpha}\|_2^2}, \frac{\epsilon}{K\|\boldsymbol{\alpha}\|_\infty}\right)\right), \tag{25}$$

*where $c > 0$ is a constant.*

## B    Proof of Theorem 1 (Recovery Guarantee for Amplitude-Based Loss Minimization)

Before proving the theorem, we present some additional auxiliary results.

### B.1 Useful Lemmas

First, based on a basic two-sided concentration bound for standard Gaussian matrices (*cf.*, Lemma 1 in Appendix A), and the well-established chaining arguments used in [5, 36], we have the following lemma, which essentially gives a two-sided Set-Restricted Eigenvalue Condition (S-REC) [5].

**Lemma 4.** [5, Lemma 4.1], [36, Lemma 2] *For $\alpha < 1$ and $\delta > 0$, if $m = \Omega\left(\frac{k}{\alpha^2} \log \frac{Lr}{\delta}\right)$, then with probability $1 - e^{-\Omega(\alpha^2 m)}$, we have for all $\mathbf{x}_1, \mathbf{x}_2 \in G(B_2^k(r))$ that*

$$(1 - \alpha)\|\mathbf{x}_1 - \mathbf{x}_2\|_2 - \delta \leq \frac{1}{\sqrt{m}}\|\mathbf{A}(\mathbf{x}_1 - \mathbf{x}_2)\|_2 \leq (1 + \alpha)\|\mathbf{x}_1 - \mathbf{x}_2\|_2 + \delta. \tag{26}$$

In addition, we have the following lemma, which is similar to Lemma 1 in Appendix A.

**Lemma 5.** [61, Lemma 4.3] *Suppose that $X_1, X_2, \ldots, X_N$ are i.i.d. standard normal random variables. Then, for $0 \leq \epsilon \leq \mu$, we have with probability $1 - e^{-\Omega(N\epsilon^2)}$ that*

$$\sqrt{N}(\mu - \epsilon) < \sqrt{\sum_{i=1}^{\lceil N/2 \rceil} |X|_{(i)}^2} < \sqrt{N}(\mu + \epsilon), \tag{27}$$

*where $\mu \geq \frac{1}{18}\sqrt{\frac{\pi}{2}}$ is a positive constant, and $|X|_{(1)} \leq |X|_{(2)} \leq \ldots \leq |X|_{(N)}$, i.e., $|X|_{(k)}$ is the $k$-th smallest entry in $|X_1|, |X_2|, \ldots, |X_N|$.*

Based on Lemma 5 and using a chaining argument as in [5, 35], we arrive at the following lemma, which can be viewed as another (one-sided) variant of the S-REC in Lemma 4. The proof of Lemma 6 follows easily from [5, 35]; for completeness, we provide an outline in Appendix D.

**Lemma 6.** *Let $\mu \geq \frac{1}{18}\sqrt{\frac{\pi}{2}}$ be the same positive constant as in Lemma 5. For $\alpha < \mu$ and $\delta > 0$, if $m = \Omega\left(\frac{k}{\alpha^2} \log \frac{Lr}{\delta}\right)$, then with probability $1 - e^{-\Omega(\alpha^2 m)}$, we have for all $\mathbf{x}_1, \mathbf{x}_2 \in G(B_2^k(r))$ that*

$$(\mu - \alpha)\|\mathbf{x}_1 - \mathbf{x}_2\|_2 - \delta \leq \min_{I \subseteq [m], |I| \geq \frac{m}{2}} \frac{1}{\sqrt{m}}\|\mathbf{A}_I(\mathbf{x}_1 - \mathbf{x}_2)\|_2, \tag{28}$$

*where for any index set $I \subseteq [m]$, $\mathbf{A}_I$ denotes the $|I| \times n$ sub-matrix of $\mathbf{A}$ that only keeps the rows indexed by $I$.*

### B.2 Proof of Theorem 1

Since $\mathbf{p} \in \text{Range}(G)$, we have

$$\|\mathbf{y} - |\mathbf{A}\mathbf{p}|\|_2^2 + \tau \geq \min_{\mathbf{w} \in \text{Range}(G)} \|\mathbf{y} - |\mathbf{A}\mathbf{w}|\|_2^2 + \tau \tag{29}$$

$$\geq \|\mathbf{y} - |\mathbf{A}\mathbf{q}|\|_2^2 \tag{30}$$

$$= \|(\mathbf{y} - |\mathbf{A}\mathbf{p}|) - (|\mathbf{A}\mathbf{q}| - |\mathbf{A}\mathbf{p}|)\|_2^2, \tag{31}$$

where (30) follows from (5). Let $S, T$ be the following index sets:

$$S := \{i \in [m] : \text{sign}(\mathbf{a}_i^T \mathbf{p}) = \text{sign}(\mathbf{a}_i^T \mathbf{q})\}, \tag{32}$$

$$T := \{i \in [m] : \text{sign}(\mathbf{a}_i^T \mathbf{p}) \neq \text{sign}(\mathbf{a}_i^T \mathbf{q})\}. \tag{33}$$

Without loss of generality, we assume that $|S| \geq \frac{m}{2}$ (otherwise, we have $|T| \geq \frac{m}{2}$ and we can derive an upper bound for $\|\mathbf{q} + \mathbf{x}\|_2$ instead of for $\|\mathbf{q} - \mathbf{x}\|_2$). Expanding the squares in (31) and rearranging, we obtain

$$\||\mathbf{A}\mathbf{p}| - |\mathbf{A}\mathbf{q}|\|_2^2 \leq 2\langle \mathbf{y} - |\mathbf{A}\mathbf{p}|, |\mathbf{A}\mathbf{q}| - |\mathbf{A}\mathbf{p}|\rangle + \tau \tag{34}$$

$$\leq 2\|\mathbf{y} - |\mathbf{A}\mathbf{p}|\|_2 \||\mathbf{A}\mathbf{q}| - |\mathbf{A}\mathbf{p}|\|_2 + \tau \tag{35}$$

$$\leq 2\|\mathbf{y} - |\mathbf{A}\mathbf{p}|\|_2 \|\mathbf{A}(\mathbf{q} - \mathbf{p})\|_2 + \tau \tag{36}$$

$$\leq 2(\|\boldsymbol{\eta}\|_2 + \|\mathbf{A}(\mathbf{p} - \mathbf{x})\|_2)\|\mathbf{A}(\mathbf{q} - \mathbf{p})\|_2 + \tau, \tag{37}$$

where (36) follows from the inequality $||a| - |b|| \leq \min\{|a - b|, |a + b|\}$ for $a, b \in \mathbb{R}$, and (37) follows from (1) and the triangle inequality. Note that $\mathbf{p} - \mathbf{x}$ is a fixed vector. From a basic concentration

bound for standard Gaussian matrices (*cf.*, Lemma 1 in Appendix A), we have with probability $1 - e^{-\Omega(m)}$ that

$$\|\mathbf{A}(\mathbf{p} - \mathbf{x})\|_2 \leq 2\sqrt{m}\|\mathbf{p} - \mathbf{x}\|_2. \tag{38}$$

In addition, setting $\alpha = \frac{1}{2}$ in Lemma 4, we obtain that if $m = \Omega\left(k \log \frac{Lr}{\delta}\right)$, then with probability $1 - e^{-\Omega(m)}$, it holds that

$$\|\mathbf{A}(\mathbf{q} - \mathbf{p})\|_2 \leq \sqrt{m}\left(\frac{3}{2}\|\mathbf{q} - \mathbf{p}\|_2 + \delta\right). \tag{39}$$

Moreover, setting $\alpha = \frac{\mu}{2}$ in Lemma 6, we have that if $m = \Omega\left(k \log \frac{Lr}{\delta}\right)$, with probability $1 - e^{-\Omega(m)}$, it holds that

$$\||\mathbf{A}\mathbf{p}| - |\mathbf{A}\mathbf{q}|\|_2^2 = \|\mathbf{A}_S(\mathbf{p} - \mathbf{q})\|_2^2 + \|\mathbf{A}_T(\mathbf{p} + \mathbf{q})\|_2^2 \tag{40}$$

$$\geq \|\mathbf{A}_S(\mathbf{p} - \mathbf{q})\|_2^2 \tag{41}$$

$$\geq m\left(\frac{\mu}{2}\|\mathbf{p} - \mathbf{q}\|_2 - \delta\right)^2 \tag{42}$$

$$\geq m\left(\frac{\mu^2}{4}\|\mathbf{p} - \mathbf{q}\|_2^2 - \mu\delta\|\mathbf{p} - \mathbf{q}\|_2\right), \tag{43}$$

where (40) follows from the definitions of $S$ and $T$, and (42) follows from Lemma 6 and our assumption $|S| \geq \frac{m}{2}$. Combining (37), (38), (39), and (43), we obtain

$$\frac{\mu^2}{4}\|\mathbf{p} - \mathbf{q}\|_2^2 - \mu\delta\|\mathbf{p} - \mathbf{q}\|_2 \leq 2\left(\frac{\|\boldsymbol{\eta}\|_2}{\sqrt{m}} + 2\|\mathbf{p} - \mathbf{x}\|_2\right)\left(\frac{3}{2}\|\mathbf{p} - \mathbf{q}\|_2 + \delta\right) + \frac{\tau}{m}, \tag{44}$$

which implies

$$\|\mathbf{p} - \mathbf{q}\|_2^2 \leq O\left(\|\mathbf{p} - \mathbf{x}\|_2 + \frac{\|\boldsymbol{\eta}\|_2}{\sqrt{m}} + \delta\right)\|\mathbf{p} - \mathbf{q}\|_2 + \left(\|\mathbf{p} - \mathbf{x}\|_2 + \frac{\|\boldsymbol{\eta}\|_2}{\sqrt{m}}\right)\delta + O\left(\frac{\tau}{m}\right). \tag{45}$$

Simplifying terms in (45), we obtain

$$\|\mathbf{p} - \mathbf{q}\|_2 \leq O\left(\|\mathbf{p} - \mathbf{x}\|_2 + \frac{\|\boldsymbol{\eta}\|_2 + \sqrt{\tau}}{\sqrt{m}} + \delta\right). \tag{46}$$

From the triangle inequality $\|\mathbf{q} - \mathbf{x}\|_2 \leq \|\mathbf{p} - \mathbf{q}\|_2 + \|\mathbf{p} - \mathbf{x}\|_2$, we obtain the desired result.

## C    Proof of Theorem 2 (Spectral Initializations with Generative Priors)

Before proving Theorem 2, we provide a simplified outline and some useful auxiliary results. In this appendix, it is particularly important to remember that $c$ and $c'$ represent small constants whose values may differ from line to line.

### C.1    Simplified Outline of the Proof

Since the full proof of Theorem 2 is rather lengthy, we first provide an outline with certain simplifying assumptions (which are non-rigorous but will be dropped subsequently). Specifically, we consider an idealized scenario in which $\boldsymbol{\eta} = \mathbf{0}$ (i.e, the noiseless setting), and $\lambda$ exactly equals its expectation $\|\mathbf{x}\|_2$. Then, we have $y_i = |\langle \mathbf{a}_i, \mathbf{x}\rangle|$, and $\mathbf{V}$ simplifies to

$$\mathbf{V}_0 = \frac{1}{m}\sum_{i=1}^m y_i \mathbf{a}_i \mathbf{a}_i^T \mathbf{1}_{\{l < |\mathbf{a}_i^T \bar{\mathbf{x}}| < u\}}. \tag{47}$$

In this simplified scenario, the mean is given by (*cf.* (13))

$$\mathbf{J} = \mathbb{E}[\mathbf{V}_0] = \|\mathbf{x}\|_2(\gamma_0 \mathbf{I}_n + \beta_0 \bar{\mathbf{x}}\bar{\mathbf{x}}^T), \tag{48}$$

and $\hat{\mathbf{x}} \in \mathcal{S}^{n-1}$ is the vector that satisfies

$$\hat{\mathbf{x}} = \arg\max_{\mathbf{w} \in \tilde{G}(\mathcal{D})} \mathbf{w}^T \mathbf{V}_0 \mathbf{w}. \tag{49}$$

On the one hand, since $\bar{\mathbf{x}} \in \tilde{G}(\mathcal{D})$, we have

$$\hat{\mathbf{x}}^T \mathbf{V}_0 \hat{\mathbf{x}} \geq \bar{\mathbf{x}}^T \mathbf{V}_0 \bar{\mathbf{x}} \tag{50}$$

$$= \bar{\mathbf{x}}^T (\mathbf{V}_0 - \mathbf{J}) \bar{\mathbf{x}} + \bar{\mathbf{x}}^T \mathbf{J} \bar{\mathbf{x}} \tag{51}$$

$$= \bar{\mathbf{x}}^T (\mathbf{V}_0 - \mathbf{J}) \bar{\mathbf{x}} + \|\mathbf{x}\|_2 (\gamma_0 + \beta_0), \tag{52}$$

where (52) uses (48). With $y_i = |\mathbf{a}_i^T \mathbf{x}|$, we have that $y_i (\mathbf{a}_i^T \bar{\mathbf{x}})^2 \mathbf{1}_{\{l < |\mathbf{a}_i^T \bar{\mathbf{x}}| < u\}}$ is upper bounded by $\|\mathbf{x}\|_2 u^3$. From (52) we can use the concentration inequality for the sum of sub-Gaussian random variables (*cf.* Lemma 2) to derive a lower bound of $\hat{\mathbf{x}}^T \mathbf{V}_0 \hat{\mathbf{x}}$.

On the other hand, we have

$$\hat{\mathbf{x}}^T \mathbf{V}_0 \hat{\mathbf{x}} = \hat{\mathbf{x}}^T (\mathbf{V}_0 - \mathbf{J}) \hat{\mathbf{x}} + \hat{\mathbf{x}}^T \mathbf{J} \hat{\mathbf{x}} \tag{53}$$

$$= \hat{\mathbf{x}}^T (\mathbf{V}_0 - \mathbf{J}) \hat{\mathbf{x}} + \|\mathbf{x}\|_2 \left( \gamma_0 + \beta_0 (\hat{\mathbf{x}}^T \bar{\mathbf{x}})^2 \right), \tag{54}$$

where we again used (48). Note that $\hat{\mathbf{x}}$ is dependent on $\mathbf{V}_0$. To upper bound (54), we construct a $\delta$-net, and write $\hat{\mathbf{x}}$ as

$$\hat{\mathbf{x}} = (\hat{\mathbf{x}} - \tilde{\mathbf{x}}) + \tilde{\mathbf{x}}, \tag{55}$$

where $\tilde{\mathbf{x}}$ is in the $\delta$-net satisfying $\|\tilde{\mathbf{x}} - \hat{\mathbf{x}}\|_2 < \delta$. For any $\mathbf{s} \in \mathbb{R}^n$, we have that $y_i (\mathbf{a}_i^T \mathbf{s})^2 \mathbf{1}_{\{l < |\mathbf{a}_i^T \bar{\mathbf{x}}| < u\}}$ is sub-exponential, with the sub-exponential norm being upper bounded by $\|\mathbf{x}\|_2 u C \|\mathbf{s}\|_2^2$, where $C$ is an absolute constant. Using a concentration inequality for the sum of sub-exponential random variables (*cf.* Lemma 3) and taking a union bound over the $\delta$-net, we obtain an upper bound for $\tilde{\mathbf{x}}^T (\mathbf{V}_0 - \mathbf{J}) \tilde{\mathbf{x}}$. Using a high-probability upper bound on $\|\mathbf{a}_i\|_2$ and the fact that $\|\tilde{\mathbf{x}} - \hat{\mathbf{x}}\|_2 < \delta$, we can control the terms $(\hat{\mathbf{x}} - \tilde{\mathbf{x}})^T (\mathbf{V}_0 - \mathbf{J}) \hat{\mathbf{x}}$ and $(\hat{\mathbf{x}} - \tilde{\mathbf{x}})^T (\mathbf{V}_0 - \mathbf{J})(\hat{\mathbf{x}} - \tilde{\mathbf{x}})$. Then, from (54), we obtain an upper bound on $\hat{\mathbf{x}}^T \mathbf{V}_0 \hat{\mathbf{x}}$. Combining the upper and lower bounds on $\hat{\mathbf{x}}^T \mathbf{V}_0 \hat{\mathbf{x}}$, we can derive the desired result.

In the full analysis in Section C.3, we additionally carefully deal with the noise terms and the case that $\lambda$ approximately equals $\|\mathbf{x}\|_2$.

## C.2 Additional Lemmas

Based on Lemma 2, we present the following simple lemma showing that $\lambda$ approximates $\|\mathbf{x}\|_2$.

**Lemma 7.** *For any fixed $c \in (0, 1)$, with probability $1 - e^{-\Omega(m)}$, we have that $\lambda$ defined in (11) satisfies*

$$1 - c < \frac{\lambda}{\|\mathbf{x}\|_2} < 1 + c. \tag{56}$$

*Proof.* Since $|\langle \mathbf{a}_i, \mathbf{x} \rangle|$ is sub-Gaussian with the sub-Gaussian norm upper bounded by $C \|\mathbf{x}\|_2$ and $\mathbb{E}[|\langle \mathbf{a}_i, \mathbf{x} \rangle|] = \sqrt{\frac{2}{\pi}} \|\mathbf{x}\|_2$, from Lemma 2, we have with probability $1 - e^{-\Omega(m)}$ that

$$\left| \frac{1}{m} \sum_{i=1}^{m} |\langle \mathbf{a}_i, \mathbf{x} \rangle| - \sqrt{\frac{2}{\pi}} \|\mathbf{x}\|_2 \right| \leq c \|\mathbf{x}\|_2. \tag{57}$$

In addition, by the Cauchy–Schwarz inequality and the upper bound for the noise term assumed in (8), we have

$$\left| \frac{1}{m} \sum_{i=1}^{m} \eta_i \right| \leq \frac{\|\boldsymbol{\eta}\|_2}{\sqrt{m}} \leq c_0 \|\mathbf{x}\|_2. \tag{58}$$

Then, using (57), (58), and the triangle inequality, we obtain

$$\left( \sqrt{\frac{2}{\pi}} - c' \right) \|\mathbf{x}\|_2 \leq \frac{1}{m} \sum_{i=1}^{m} y_i = \frac{1}{m} \sum_{i=1}^{m} (|\langle \mathbf{a}_i, \mathbf{x} \rangle| + \eta_i) \leq \left( \sqrt{\frac{2}{\pi}} + c' \right) \|\mathbf{x}\|_2. \tag{59}$$

Therefore, we obtain

$$(1 - c) \|\mathbf{x}\|_2 < \lambda = \sqrt{\frac{\pi}{2}} \left( \frac{1}{m} \sum_{i=1}^{m} y_i \right) < (1 + c) \|\mathbf{x}\|_2 \tag{60}$$

for a choice of $c$ possibly different from that above, but still arbitrarily small. $\square$

Next, we present the following lemma concerning the expectation of the truncated empirical matrix.

**Lemma 8.** *Suppose that* $\mathbf{a} \sim \mathcal{N}(\mathbf{0}, \mathbf{I}_n)$. *For any* $\mathbf{s} \in \mathbb{R}^n$ *and* $\eta \in \mathbb{R}$, *choosing* $\alpha_1, \alpha_2 \in \mathbb{R}$ *such that* $1 + \frac{\eta}{\|\mathbf{s}\|_2} < \alpha_1 < \alpha_2$ *and setting* $y = |\langle \mathbf{a}, \mathbf{s} \rangle| + \eta$, *we have*

$$\mathbb{E}\left[y\mathbf{a}\mathbf{a}^T \mathbf{1}_{\{\alpha_1\|\mathbf{s}\|_2 < y < \alpha_2\|\mathbf{s}\|_2\}}\right] = \|\mathbf{s}\|_2 \left(\gamma_1 \mathbf{I}_n + \beta_1 \bar{\mathbf{s}}\bar{\mathbf{s}}^T\right) + \eta(\check{\gamma}_1 \mathbf{I}_n + \check{\beta}_1 \bar{\mathbf{s}}\bar{\mathbf{s}}^T), \tag{61}$$

*where* $\bar{\mathbf{s}} := \frac{\mathbf{s}}{\|\mathbf{s}\|_2}$, *and for* $g \sim \mathcal{N}(0,1)$ *and* $\phi_1(x) = \mathbf{1}_{\{\alpha_1 - \frac{\eta}{\|\mathbf{s}\|_2} < |x| < \alpha_2 - \frac{\eta}{\|\mathbf{s}\|_2}\}}$, *we define* $\gamma_1 = \mathbb{E}[|g|\phi_1(g)]$, $\beta_1 = \mathbb{E}[|g|^3\phi_1(g)] - \gamma_1$, $\check{\gamma}_1 = \mathbb{E}[\phi_1(g)]$ *and* $\check{\beta}_1 = \mathbb{E}[g^2\phi_1(g)] - \check{\gamma}_1$. *(Note that from the assumption that* $1 + \frac{\eta}{\|\mathbf{s}\|_2} < \alpha_1 < \alpha_2$, *we have* $\beta_1 > 0$ *and* $\check{\beta}_1 > 0$.)

*Proof.* Let $g = \langle \mathbf{a}, \bar{\mathbf{s}} \rangle \sim \mathcal{N}(0,1)$. We have

$$y\mathbf{a}\mathbf{a}^T \mathbf{1}_{\{\alpha_1\|\mathbf{s}\|_2 < y < \alpha_2\|\mathbf{s}\|_2\}} = (\|\mathbf{s}\|_2|\mathbf{a}^T\bar{\mathbf{s}}| + \eta)\mathbf{a}\mathbf{a}^T \mathbf{1}_{\{\alpha_1 - \frac{\eta}{\|\mathbf{s}\|_2} < |\langle \mathbf{a}, \bar{\mathbf{s}} \rangle| < \alpha_2 - \frac{\eta}{\|\mathbf{s}\|_2}\}} \tag{62}$$

$$= (\|\mathbf{s}\|_2|g| + \eta)\phi_1(g)\mathbf{a}\mathbf{a}^T. \tag{63}$$

For any $i \in [n]$, we have $\text{Cov}[a_i, g] = \bar{s}_i$, and thus $a_i$ can be written as

$$a_i = \bar{s}_i g + t_i, \tag{64}$$

where $t_i \sim \mathcal{N}(0, 1 - \bar{s}_i^2)$ is independent of $g$. Then, we have

$$\mathbb{E}\left[|g|\phi_1(g)a_i^2\right] = \mathbb{E}\left[|g|\phi_1(g)(\bar{s}_i g + t_i)^2\right] \tag{65}$$

$$= \bar{s}_i^2 \mathbb{E}\left[|g|^3\phi_1(g)\right] + \left(1 - \bar{s}_i^2\right)\mathbb{E}[|g|\phi_1(g)] \tag{66}$$

$$= \gamma_1 + \beta_1 \bar{s}_i^2, \tag{67}$$

and similarly,

$$\mathbb{E}\left[\phi_1(g)a_i^2\right] = \mathbb{E}\left[\phi_1(g)(\bar{s}_i g + t_i)^2\right] \tag{68}$$

$$= \bar{s}_i^2 \mathbb{E}\left[g^2\phi_1(g)\right] + \left(1 - \bar{s}_i^2\right)\mathbb{E}[\phi_1(g)] \tag{69}$$

$$= \check{\gamma}_1 + \check{\beta}_1 \bar{s}_i^2. \tag{70}$$

Moreover, for $1 \leq i \neq j \leq n$, we have $0 = \mathbb{E}[a_i a_j] = \mathbb{E}[(\bar{s}_i g + t_i)(\bar{s}_j g + t_j)] = \bar{s}_i \bar{s}_j + \mathbb{E}[t_i t_j]$, which gives $\mathbb{E}[t_i t_j] = -\bar{s}_i \bar{s}_j$. Then, similarly to (67) and (70), we have

$$\mathbb{E}\left[|g|\phi_1(g)a_i a_j\right] = \mathbb{E}\left[|g|\phi_1(g)(\bar{s}_i g + t_i)(\bar{s}_j g + t_j)\right] = \beta_1 \bar{s}_i \bar{s}_j, \tag{71}$$

and

$$\mathbb{E}\left[\phi_1(g)a_i a_j\right] = \mathbb{E}\left[\phi_1(g)(\bar{s}_i g + t_i)(\bar{s}_j g + t_j)\right] = \check{\beta}_1 \bar{s}_i \bar{s}_j. \tag{72}$$

Combining (63), (67), (70), (71), (72), we obtain the desired result. $\square$

### C.3 Proof of Theorem 2

Let $\mathcal{E}$ be the event that

$$1 - c \leq \frac{\lambda}{\|\mathbf{x}\|_2} \leq 1 + c. \tag{73}$$

From Lemma 7, we know that $\mathcal{E}$ occurs with probability $1 - e^{-\Omega(m)}$. Throughout the following, we assume that $\mathcal{E}$ holds, and the relevant probabilities are conditioned accordingly.

**Lower bounding** $\hat{\mathbf{x}}^T \mathbf{V} \hat{\mathbf{x}}$**:** Since $\bar{\mathbf{x}} \in \tilde{G}(\mathcal{D})$, and $\hat{\mathbf{x}}$ is a solution of (14), we have

$$\hat{\mathbf{x}}^T \mathbf{V} \hat{\mathbf{x}} \geq \bar{\mathbf{x}}^T \mathbf{V} \bar{\mathbf{x}} \tag{74}$$

$$= \frac{1}{m} \sum_{i=1}^{m} y_i \left(\mathbf{a}_i^T \bar{\mathbf{x}}\right)^2 \mathbf{1}_{\{l\lambda < y_i < u\lambda\}} \tag{75}$$

$$= \frac{\|\mathbf{x}\|_2}{m} \sum_{i=1}^{m} |\mathbf{a}_i^T \bar{\mathbf{x}}|^3 \mathbf{1}_{\{l\lambda < y_i < u\lambda\}} + \frac{1}{m} \sum_{i=1}^{m} \eta_i \left(\mathbf{a}_i^T \bar{\mathbf{x}}\right)^2 \mathbf{1}_{\{l\lambda < y_i < u\lambda\}} \tag{76}$$

$$\geq \frac{\|\mathbf{x}\|_2}{m} \sum_{i=1}^{m} |\mathbf{a}_i^T \bar{\mathbf{x}}|^3 \mathbf{1}_{\{l(1+c')\|\mathbf{x}\|_2 < y_i < u(1-c')\|\mathbf{x}\|_2\}}$$

$$- \frac{1}{m} \sum_{i=1}^{m} |\eta_i| \left(\mathbf{a}_i^T \bar{\mathbf{x}}\right)^2 \mathbf{1}_{\{l(1-c')\|\mathbf{x}\|_2 < y_i < u(1+c')\|\mathbf{x}\|_2\}} \tag{77}$$

$$= \frac{\|\mathbf{x}\|_2}{m} \sum_{i=1}^{m} |\mathbf{a}_i^T \bar{\mathbf{x}}|^3 \mathbf{1}_{\{l(1+c') - \frac{\eta_i}{\|\mathbf{x}\|_2} < |\mathbf{a}_i^T \bar{\mathbf{x}}| < u(1-c') - \frac{\eta_i}{\|\mathbf{x}\|_2}\}}$$

$$- \frac{1}{m} \sum_{i=1}^{m} |\eta_i| \left(\mathbf{a}_i^T \bar{\mathbf{x}}\right)^2 \mathbf{1}_{\{l(1-c') - \frac{\eta_i}{\|\mathbf{x}\|_2} < |\mathbf{a}_i^T \bar{\mathbf{x}}| < u(1+c') - \frac{\eta_i}{\|\mathbf{x}\|_2}\}} \tag{78}$$

$$\geq \frac{\|\mathbf{x}\|_2}{m} \sum_{i=1}^{m} |\mathbf{a}_i^T \bar{\mathbf{x}}|^3 \mathbf{1}_{\{l(1+c) < |\mathbf{a}_i^T \bar{\mathbf{x}}| < u(1-c)\}} - \frac{1}{m} \sum_{i=1}^{m} |\eta_i| \left(\mathbf{a}_i^T \bar{\mathbf{x}}\right)^2 \mathbf{1}_{\{l(1-c) < |\mathbf{a}_i^T \bar{\mathbf{x}}| < u(1+c)\}}, \tag{79}$$

where we use (73) in (77), we use $y_i = |\mathbf{a}_i^T \mathbf{x}| + \eta_i$ in (76) and (78), and we use (9) as well as $u > l > 1 + c_1$ to derive (79). We aim to bound the two terms in (79). Let

$$\mathbf{U} = \frac{1}{m} \sum_{i=1}^{m} |\mathbf{a}_i^T \mathbf{x}| \mathbf{a}_i \mathbf{a}_i^T \mathbf{1}_{\{l(1+c) < |\mathbf{a}_i^T \bar{\mathbf{x}}| < u(1-c)\}}, \tag{80}$$

$$\mathbf{W} = \frac{1}{m} \sum_{i=1}^{m} |\mathbf{a}_i^T \mathbf{x}| \mathbf{a}_i \mathbf{a}_i^T \mathbf{1}_{\{l(1-c) < |\mathbf{a}_i^T \bar{\mathbf{x}}| < u(1+c)\}}, \tag{81}$$

$$\check{\mathbf{U}} = \frac{1}{m} \sum_{i=1}^{m} |\eta_i| \mathbf{a}_i \mathbf{a}_i^T \mathbf{1}_{\{l(1+c) < |\mathbf{a}_i^T \bar{\mathbf{x}}| < u(1-c)\}}, \tag{82}$$

$$\check{\mathbf{W}} = \frac{1}{m} \sum_{i=1}^{m} |\eta_i| \mathbf{a}_i \mathbf{a}_i^T \mathbf{1}_{\{l(1-c) < |\mathbf{a}_i^T \bar{\mathbf{x}}| < u(1+c)\}}. \tag{83}$$

According to the proof of Lemma 8, we have

$$\mathbb{E}[\mathbf{U}] = \|\mathbf{x}\|_2 \left(\gamma \mathbf{I}_n + \beta \bar{\mathbf{x}} \bar{\mathbf{x}}^T\right), \tag{84}$$

$$\mathbb{E}[\mathbf{W}] = \|\mathbf{x}\|_2 \left(\gamma' \mathbf{I}_n + \beta' \bar{\mathbf{x}} \bar{\mathbf{x}}^T\right), \tag{85}$$

$$\mathbb{E}[\check{\mathbf{U}}] = \bar{\eta} \left(\check{\gamma} \mathbf{I}_n + \check{\beta} \bar{\mathbf{x}} \bar{\mathbf{x}}^T\right), \tag{86}$$

$$\mathbb{E}[\check{\mathbf{W}}] = \bar{\eta} \left(\check{\gamma}' \mathbf{I}_n + \check{\beta}' \bar{\mathbf{x}} \bar{\mathbf{x}}^T\right). \tag{87}$$

where for $g \sim \mathcal{N}(0, 1)$ and

$$\phi(x) := \mathbf{1}_{\{l(1+c) < |x| < u(1-c)\}}, \tag{88}$$

$$\psi(x) := \mathbf{1}_{\{l(1-c) < |x| < u(1+c)\}}, \tag{89}$$

we set $\gamma = \mathbb{E}[|g|\phi(g)]$, $\beta = \mathbb{E}[|g|^3 \phi(g)] - \gamma$, $\gamma' = \mathbb{E}[|g|\psi(g)]$, $\beta' = \mathbb{E}[|g|^3 \psi(g)] - \gamma'$, $\bar{\eta} = \frac{1}{m} \sum_{i=1}^{m} |\eta_i|$, $\check{\gamma} = \mathbb{E}[\phi(g)]$, $\check{\beta} = \mathbb{E}[g^2 \phi(g)] - \check{\gamma}$, $\check{\gamma}' = \mathbb{E}[\psi(g)]$, and $\check{\beta}' = \mathbb{E}[g^2 \psi(g)] - \check{\gamma}'$. Then, in (79), we have

$$\frac{\|\mathbf{x}\|_2}{m} \sum_{i=1}^{m} |\mathbf{a}_i^T \bar{\mathbf{x}}|^3 \mathbf{1}_{\{l(1+c) < |\mathbf{a}_i^T \bar{\mathbf{x}}| < u(1-c)\}}$$

$$= \bar{\mathbf{x}}^T \mathbf{U} \bar{\mathbf{x}} \tag{90}$$

$$= \bar{\mathbf{x}}^T(\mathbf{U} - \mathbb{E}[\mathbf{U}])\bar{\mathbf{x}} + \bar{\mathbf{x}}^T\mathbb{E}[\mathbf{U}]\bar{\mathbf{x}} \tag{91}$$

$$= \frac{1}{m}\sum_{i=1}^{m}\left(|\mathbf{a}_i^T\mathbf{x}|(\mathbf{a}_i^T\bar{\mathbf{x}})^2\phi\left(\mathbf{a}_i^T\bar{\mathbf{x}}\right) - \mathbb{E}\left[|\mathbf{a}_i^T\mathbf{x}|(\mathbf{a}_i^T\bar{\mathbf{x}})^2\phi\left(\mathbf{a}_i^T\bar{\mathbf{x}}\right)\right]\right) + \|\mathbf{x}\|_2(\gamma + \beta), \tag{92}$$

where we use (84), (88), and $\|\bar{\mathbf{x}}\|_2 = 1$ to obtain (92). Note that $|\mathbf{a}_i^T\mathbf{x}|(\mathbf{a}_i^T\bar{\mathbf{x}})^2\phi\left(\mathbf{a}_i^T\bar{\mathbf{x}}\right) = |\mathbf{a}_i^T\mathbf{x}|(\mathbf{a}_i^T\bar{\mathbf{x}})^2\mathbf{1}_{\{l(1+c)<|\mathbf{a}_i^T\bar{\mathbf{x}}|<u(1-c)\}} \leq u^3(1-c)^3\|\mathbf{x}\|_2 = O(\|\mathbf{x}\|_2)$. Then, from Lemma 2, we have with probability $1 - e^{-\Omega(m)}$ that

$$\left|\frac{1}{m}\sum_{i=1}^{m}\left(|\mathbf{a}_i^T\mathbf{x}|(\mathbf{a}_i^T\bar{\mathbf{x}})^2\phi\left(\mathbf{a}_i^T\bar{\mathbf{x}}\right) - \mathbb{E}\left[|\mathbf{a}_i^T\mathbf{x}|(\mathbf{a}_i^T\bar{\mathbf{x}})^2\phi\left(\mathbf{a}_i^T\bar{\mathbf{x}}\right)\right]\right)\right| \leq c\|\mathbf{x}\|_2. \tag{93}$$

In addition, we have

$$\frac{1}{m}\sum_{i=1}^{m}|\eta_i|\left(\mathbf{a}_i^T\bar{\mathbf{x}}\right)^2\mathbf{1}_{\{l(1-c)<|\mathbf{a}_i^T\bar{\mathbf{x}}|<u(1+c)\}}$$

$$= \bar{\mathbf{x}}^T\check{\mathbf{W}}\bar{\mathbf{x}} \tag{94}$$

$$= \bar{\mathbf{x}}^T(\check{\mathbf{W}} - \mathbb{E}[\check{\mathbf{W}}])\bar{\mathbf{x}} + \bar{\mathbf{x}}^T\mathbb{E}[\check{\mathbf{W}}]\bar{\mathbf{x}} \tag{95}$$

$$= \frac{1}{m}\sum_{i=1}^{m}|\eta_i|\left(\left(\mathbf{a}_i^T\bar{\mathbf{x}}\right)^2\psi\left(\mathbf{a}_i^T\bar{\mathbf{x}}\right) - \mathbb{E}\left[\left(\mathbf{a}_i^T\bar{\mathbf{x}}\right)^2\psi\left(\mathbf{a}_i^T\bar{\mathbf{x}}\right)\right]\right) + \bar{\eta}(\check{\gamma}' + \check{\beta}') \tag{96}$$

$$\leq \frac{1}{m}\sum_{i=1}^{m}|\eta_i|\left(\left(\mathbf{a}_i^T\bar{\mathbf{x}}\right)^2\psi\left(\mathbf{a}_i^T\bar{\mathbf{x}}\right) - \mathbb{E}\left[\left(\mathbf{a}_i^T\bar{\mathbf{x}}\right)^2\psi\left(\mathbf{a}_i^T\bar{\mathbf{x}}\right)\right]\right) + c_0\|\mathbf{x}\|_2(\check{\gamma}' + \check{\beta}'), \tag{97}$$

where we use $\bar{\eta} = \frac{1}{m}\sum_{i=1}^{m}|\eta_i| \leq \frac{\|\boldsymbol{\eta}\|_2}{\sqrt{m}} \leq c_0\|\mathbf{x}\|_2$ (see (8)) in (97). From the definition of $\psi$ in (89), we have $\left(\mathbf{a}_i^T\bar{\mathbf{x}}\right)^2\psi\left(\mathbf{a}_i^T\bar{\mathbf{x}}\right) \leq u^2(1+c)^2 = O(1)$. Setting $\alpha_i = \frac{|\eta_i|}{m}$ and $X_i = \left(\mathbf{a}_i^T\bar{\mathbf{x}}\right)^2\psi\left(\mathbf{a}_i^T\bar{\mathbf{x}}\right) - \mathbb{E}\left[\left(\mathbf{a}_i^T\bar{\mathbf{x}}\right)^2\psi\left(\mathbf{a}_i^T\bar{\mathbf{x}}\right)\right]$ in Lemma 2, we obtain with probability $1 - e^{-\Omega(m)}$ that

$$\left|\frac{1}{m}\sum_{i=1}^{m}|\eta_i|\left(\left(\mathbf{a}_i^T\bar{\mathbf{x}}\right)^2\psi\left(\mathbf{a}_i^T\bar{\mathbf{x}}\right) - \mathbb{E}\left[\left(\mathbf{a}_i^T\bar{\mathbf{x}}\right)^2\psi\left(\mathbf{a}_i^T\bar{\mathbf{x}}\right)\right]\right)\right| \leq \frac{\|\boldsymbol{\eta}\|_2}{\sqrt{m}} \leq c_0\|\mathbf{x}\|_2. \tag{98}$$

Combining (79), (92), (93), (97) and (98), we have with probability $1 - e^{-\Omega(m)}$ that

$$\hat{\mathbf{x}}^T\mathbf{V}\hat{\mathbf{x}} \geq (\gamma + \beta - c)\|\mathbf{x}\|_2. \tag{99}$$

**Upper bounding $\hat{\mathbf{x}}^T\mathbf{V}\hat{\mathbf{x}}$:** We have

$$\hat{\mathbf{x}}^T\mathbf{V}\hat{\mathbf{x}} = \frac{1}{m}\sum_{i=1}^{m}|\mathbf{a}_i^T\mathbf{x}|(\mathbf{a}_i^T\hat{\mathbf{x}})^2\mathbf{1}_{\{l\lambda<y_i<u\lambda\}} + \frac{1}{m}\sum_{i=1}^{m}\eta_i(\mathbf{a}_i^T\hat{\mathbf{x}})^2\mathbf{1}_{\{l\lambda<y_i<u\lambda\}} \tag{100}$$

$$\leq \frac{1}{m}\sum_{i=1}^{m}|\mathbf{a}_i^T\mathbf{x}|(\mathbf{a}_i^T\hat{\mathbf{x}})^2\mathbf{1}_{\{l(1-c)<|\mathbf{a}_i^T\bar{\mathbf{x}}|<u(1+c)\}} + \frac{1}{m}\sum_{i=1}^{m}|\eta_i|(\mathbf{a}_i^T\hat{\mathbf{x}})^2\mathbf{1}_{\{l(1-c)<|\mathbf{a}_i^T\bar{\mathbf{x}}|<u(1+c)\}} \tag{101}$$

$$= \hat{\mathbf{x}}^T\mathbf{W}\hat{\mathbf{x}} + \hat{\mathbf{x}}^T\check{\mathbf{W}}\hat{\mathbf{x}} \tag{102}$$

$$= \hat{\mathbf{x}}^T(\mathbf{W} - \mathbb{E}[\mathbf{W}])\hat{\mathbf{x}} + \hat{\mathbf{x}}^T(\check{\mathbf{W}} - \mathbb{E}[\check{\mathbf{W}}])\hat{\mathbf{x}} + \|\mathbf{x}\|_2\left(\gamma' + \beta'\left(\hat{\mathbf{x}}^T\bar{\mathbf{x}}\right)^2\right) + \bar{\eta}(\check{\gamma}' + \check{\beta}'\left(\hat{\mathbf{x}}^T\bar{\mathbf{x}}\right)^2) \tag{103}$$

$$\leq \hat{\mathbf{x}}^T(\mathbf{W} - \mathbb{E}[\mathbf{W}])\hat{\mathbf{x}} + \hat{\mathbf{x}}^T(\check{\mathbf{W}} - \mathbb{E}[\check{\mathbf{W}}])\hat{\mathbf{x}} + \|\mathbf{x}\|_2\left(\gamma' + \beta'\left(\hat{\mathbf{x}}^T\bar{\mathbf{x}}\right)^2\right) + c\|\mathbf{x}\|_2, \tag{104}$$

where (101) is derived similarly to (79), (102) comes from the definitions of $\mathbf{W}$ and $\check{\mathbf{W}}$ in (81) and (83), (103) follows from (85) and (87), and (104) is deduced from $\bar{\eta} \leq c_0\|\mathbf{x}\|_2$.

We aim to provide an upper bound for (104). For $\delta > 0$, let $M$ be a $(\delta/\tilde{L})$-net of $\mathcal{D} \subseteq B_2^k(r)$. There exists such a net with [60, Lemma 5.2]

$$\log|M| \leq k\log\frac{4\tilde{L}r}{\delta}. \tag{105}$$

By the $\tilde{L}$-Lipschitz continuity of $\tilde{G}$, we have that $\tilde{G}(M)$ is a $\delta$-net of $\tilde{G}(\mathcal{D})$. We write

$$\hat{\mathbf{x}} = (\hat{\mathbf{x}} - \tilde{\mathbf{x}}) + \tilde{\mathbf{x}}, \tag{106}$$

where $\tilde{\mathbf{x}} \in \tilde{G}(M)$ with $\|\hat{\mathbf{x}} - \tilde{\mathbf{x}}\|_2 \leq \delta$. Then, we have

$$\hat{\mathbf{x}}^T (\mathbf{W} - \mathbb{E}[\mathbf{W}])\hat{\mathbf{x}} = \tilde{\mathbf{x}}^T(\mathbf{W}-\mathbb{E}[\mathbf{W}])\tilde{\mathbf{x}} + 2(\hat{\mathbf{x}}-\tilde{\mathbf{x}})^T(\mathbf{W}-\mathbb{E}[\mathbf{W}])\tilde{\mathbf{x}} + (\hat{\mathbf{x}}-\tilde{\mathbf{x}})^T(\mathbf{W}-\mathbb{E}[\mathbf{W}])(\hat{\mathbf{x}}-\tilde{\mathbf{x}}). \tag{107}$$

We bound the three terms in (107) separately.

- The first term: For any $\mathbf{s} \in \tilde{G}(M)$, we have from (81) that

$$\mathbf{s}^T(\mathbf{W}-\mathbb{E}[\mathbf{W}])\mathbf{s} = \frac{1}{m}\sum_{i=1}^m \left( |\mathbf{a}_i^T\mathbf{x}| \langle \mathbf{a}_i, \mathbf{s}\rangle^2 \mathbf{1}_{\{l(1-c)<|\mathbf{a}_i^T\bar{\mathbf{x}}|<u(1+c)\}} - \mathbf{s}^T\mathbb{E}[\mathbf{W}]\mathbf{s} \right). \tag{108}$$

Note that for $\mathbf{s} \in \tilde{G}(M)$, $\|\mathbf{s}\|_2 = 1$. Then, $|\mathbf{a}_i^T\mathbf{x}|\langle \mathbf{a}_i, \mathbf{s}\rangle^2 \mathbf{1}_{\{l(1-c)<|\mathbf{a}_i^T\bar{\mathbf{x}}|<u(1+c)\}}$ is sub-exponential with the sub-exponential norm upper bounded by $Cu(1+c)\|\mathbf{x}\|_2$. From Lemma 3, we have with probability at least $1 - e^{-\Omega(m\epsilon^2)}$ that

$$\left| \mathbf{s}^T(\mathbf{W}-\mathbb{E}[\mathbf{W}])\mathbf{s} \right| \leq \epsilon\|\mathbf{x}\|_2. \tag{109}$$

Taking a union bound over all $\tilde{G}(M)$, we obtain that when $m = \Omega\left(\frac{k}{\epsilon^2}\log\frac{\tilde{L}r}{\delta}\right)$, with probability $1 - e^{-\Omega(m\epsilon^2)}$, for *all* $\mathbf{s} \in \tilde{G}(M)$,

$$\left| \mathbf{s}^T(\mathbf{W}-\mathbb{E}[\mathbf{W}])\mathbf{s} \right| \leq \epsilon\|\mathbf{x}\|_2. \tag{110}$$

Hence, since $\tilde{\mathbf{x}} \in \tilde{G}(M)$, setting $\epsilon$ to be a sufficiently small absolute constant, we obtain that when $m = \Omega\left(k\log\frac{\tilde{L}r}{\delta}\right)$, with probability $1 - e^{-\Omega(m)}$,

$$\left| \tilde{\mathbf{x}}^T(\mathbf{W}-\mathbb{E}[\mathbf{W}])\tilde{\mathbf{x}} \right| \leq c\|\mathbf{x}\|_2. \tag{111}$$

- The second term: From Lemma 2 and a union bound over $[m]$, we have with probability $1 - me^{-\Omega(n)}$ that

$$\max_{i\in[m]} \|\mathbf{a}_i\|_2 \leq \sqrt{2n}. \tag{112}$$

Conditioned on the event in (112), we have

$$(\hat{\mathbf{x}} - \tilde{\mathbf{x}})^T(\mathbf{W}-\mathbb{E}[\mathbf{W}])\tilde{\mathbf{x}}$$
$$= \frac{1}{m}\sum_{i=1}^m \left( |\mathbf{a}_i^T\mathbf{x}|(\mathbf{a}_i^T\tilde{\mathbf{x}})(\mathbf{a}_i^T(\hat{\mathbf{x}}-\tilde{\mathbf{x}}))\mathbf{1}_{\{l(1-c)<|\mathbf{a}_i^T\bar{\mathbf{x}}|<u(1+c)\}} - (\hat{\mathbf{x}}-\tilde{\mathbf{x}})^T\mathbb{E}[\mathbf{W}]\tilde{\mathbf{x}} \right) \tag{113}$$
$$= \frac{1}{m}\sum_{i=1}^m |\mathbf{a}_i^T\mathbf{x}|(\mathbf{a}_i^T\tilde{\mathbf{x}})(\mathbf{a}_i^T(\hat{\mathbf{x}}-\tilde{\mathbf{x}}))\mathbf{1}_{\{l(1-c)<|\mathbf{a}_i^T\bar{\mathbf{x}}|<u(1+c)\}}$$
$$\qquad - \|\mathbf{x}\|_2\left( \gamma'(\hat{\mathbf{x}}-\tilde{\mathbf{x}})^T\tilde{\mathbf{x}} + \beta'((\hat{\mathbf{x}}-\tilde{\mathbf{x}})^T\bar{\mathbf{x}})(\tilde{\mathbf{x}}^T\bar{\mathbf{x}}) \right) \tag{114}$$
$$\leq (2n(1+c)u + \gamma' + \beta')\|\mathbf{x}\|_2\delta, \tag{115}$$

where we use (112), $\|\hat{\mathbf{x}} - \tilde{\mathbf{x}}\|_2 \leq \delta$, $\|\tilde{\mathbf{x}}\|_2 = \|\bar{\mathbf{x}}\|_2 = 1$, and the standard inequality $|\mathbf{a}^T\mathbf{b}| \leq \|\mathbf{a}\|_2\|\mathbf{b}\|_2$ (for any $\mathbf{a}$ and $\mathbf{b}$) in (115). Therefore, we obtain with probability $1 - e^{-\Omega(n)}$ that

$$\left| 2(\hat{\mathbf{x}}-\tilde{\mathbf{x}})^T(\mathbf{W}-\mathbb{E}[\mathbf{W}])\tilde{\mathbf{x}} \right| \leq 2(2n(1+c)u + \gamma' + \beta')\|\mathbf{x}\|_2\delta. \tag{116}$$

- The third term: By an argument similar to (116), we have with probability $1 - e^{-\Omega(n)}$ that

$$\left| (\hat{\mathbf{x}}-\tilde{\mathbf{x}})^T(\mathbf{W}-\mathbb{E}[\mathbf{W}])(\hat{\mathbf{x}}-\tilde{\mathbf{x}}) \right| \leq (2n(1+c)u + \gamma' + \beta')\|\mathbf{x}\|_2\delta^2. \tag{117}$$

Combining (107), (111), (116), and (117), and setting $\delta = \frac{c}{n}$, we obtain that when $m = \Omega(k\log(\tilde{L}nr))$, with probability $1 - e^{-\Omega(m)}$, it holds that

$$\left| \hat{\mathbf{x}}^T(\mathbf{W}-\mathbb{E}[\mathbf{W}])\hat{\mathbf{x}} \right| \leq c\|\mathbf{x}\|_2. \tag{118}$$

Next, we provide an upper bound for $\hat{\mathbf{x}}^T(\check{\mathbf{W}} - \mathbb{E}[\check{\mathbf{W}}])\hat{\mathbf{x}}$. The strategy is mostly similar to that for upper bounding $\hat{\mathbf{x}}^T(\mathbf{W} - \mathbb{E}[\mathbf{W}])\hat{\mathbf{x}}$ in (104), but we provide the details below for completeness. Similarly to (107), we have

$$\hat{\mathbf{x}}^T(\check{\mathbf{W}} - \mathbb{E}[\check{\mathbf{W}}])\hat{\mathbf{x}} = \tilde{\mathbf{x}}^T(\check{\mathbf{W}} - \mathbb{E}[\check{\mathbf{W}}])\tilde{\mathbf{x}} + 2(\hat{\mathbf{x}} - \tilde{\mathbf{x}})^T(\check{\mathbf{W}} - \mathbb{E}[\check{\mathbf{W}}])\tilde{\mathbf{x}} + (\hat{\mathbf{x}} - \tilde{\mathbf{x}})^T(\check{\mathbf{W}} - \mathbb{E}[\check{\mathbf{W}}])(\hat{\mathbf{x}} - \tilde{\mathbf{x}}). \tag{119}$$

We bound the three terms in (119) separately.

- The first term: From the definition of $\check{\mathbf{W}}$ in (83), we have for any $\mathbf{s} \in \tilde{G}(M)$ that

$$\mathbf{s}^T(\check{\mathbf{W}} - \mathbb{E}[\check{\mathbf{W}}])\mathbf{s} = \frac{1}{m}\sum_{i=1}^{m}|\eta_i|(\mathbf{a}_i^T\mathbf{s})^2\mathbf{1}_{\{l(1-c)<|\mathbf{a}_i^T\bar{\mathbf{x}}|<u(1+c)\}} - \mathbf{s}^T\mathbb{E}[\check{\mathbf{W}}]\mathbf{s}. \tag{120}$$

Note that $\|\mathbf{s}\|_2 = 1$. For any $i \in [m]$, $(\mathbf{a}_i^T\mathbf{s})^2\mathbf{1}_{\{l(1-c)<|\mathbf{a}_i^T\bar{\mathbf{x}}|<u(1+c)\}}$ is sub-exponential with the sub-exponential norm upper bounded by an absolute constant $C$. Then, from Lemma 3 (see also [21, Theorem 3.1]), we obtain that for $t > 2$, with probability $1 - e^{-t}$, it holds that

$$\left|\mathbf{s}^T(\check{\mathbf{W}} - \mathbb{E}[\check{\mathbf{W}}])\mathbf{s}\right| \leq C\left(\frac{\|\boldsymbol{\eta}\|_2}{m}\sqrt{t} + \frac{\|\boldsymbol{\eta}\|_\infty}{m}t\right). \tag{121}$$

Setting $t = m$ in (121), and from $\frac{\|\boldsymbol{\eta}\|_2}{\sqrt{m}} \leq c_0\|\mathbf{x}\|_2$ (cf. (8)) and $\|\boldsymbol{\eta}\|_\infty \leq c_1\|\mathbf{x}\|_2$ (cf. (9)), we obtain with probability $1 - e^{-m}$ that

$$\left|\mathbf{s}^T(\check{\mathbf{W}} - \mathbb{E}[\check{\mathbf{W}}])\mathbf{s}\right| \leq c\|\mathbf{x}\|_2. \tag{122}$$

Taking a union bound over all $\tilde{G}(M)$, we obtain that when $m = \Omega\left(k\log\frac{\tilde{L}r}{\delta}\right)$, with probability $1 - e^{-\Omega(m\epsilon^2)}$, for *all* $\mathbf{s} \in \tilde{G}(M)$,

$$\left|\mathbf{s}^T(\check{\mathbf{W}} - \mathbb{E}[\check{\mathbf{W}}])\mathbf{s}\right| \leq c\|\mathbf{x}\|_2. \tag{123}$$

Hence, since $\tilde{\mathbf{x}} \in \tilde{G}(M)$, we have

$$\left|\tilde{\mathbf{x}}^T(\check{\mathbf{W}} - \mathbb{E}[\check{\mathbf{W}}])\tilde{\mathbf{x}}\right| \leq c\|\mathbf{x}\|_2. \tag{124}$$

- The second term: Conditioned on the event in (112), we have

$$(\hat{\mathbf{x}} - \tilde{\mathbf{x}})^T(\check{\mathbf{W}} - \mathbb{E}[\check{\mathbf{W}}])\tilde{\mathbf{x}}$$
$$= \frac{1}{m}\sum_{i=1}^{m}|\eta_i|\left(\mathbf{a}_i^T(\hat{\mathbf{x}} - \tilde{\mathbf{x}})\right)(\mathbf{a}_i^T\tilde{\mathbf{x}})\mathbf{1}_{\{l(1-c)<|\mathbf{a}_i^T\bar{\mathbf{x}}|<u(1+c)\}}$$
$$\qquad - \bar{\eta}(\check{\gamma}'(\hat{\mathbf{x}} - \tilde{\mathbf{x}})^T\tilde{\mathbf{x}} + \check{\beta}'((\hat{\mathbf{x}} - \tilde{\mathbf{x}})^T\bar{\mathbf{x}})(\tilde{\mathbf{x}}^T\bar{\mathbf{x}})) \tag{125}$$
$$\leq \bar{\eta}\delta(2n + \check{\gamma}' + \check{\beta}'). \tag{126}$$

Therefore, we obtain with probability $1 - e^{-\Omega(n)}$ that

$$\left|2(\hat{\mathbf{x}} - \tilde{\mathbf{x}})^T(\check{\mathbf{W}} - \mathbb{E}[\check{\mathbf{W}}])\tilde{\mathbf{x}}\right| \leq 2\bar{\eta}\delta(2n + \check{\gamma}' + \check{\beta}'). \tag{127}$$

- The third term: By an argument similar to (127), we have with probability $1 - e^{-\Omega(n)}$ that

$$\left|(\hat{\mathbf{x}} - \tilde{\mathbf{x}})^T(\check{\mathbf{W}} - \mathbb{E}[\check{\mathbf{W}}])(\hat{\mathbf{x}} - \tilde{\mathbf{x}})\right| \leq (2n(1+c)u + \gamma' + \beta')\bar{\eta}\delta^2. \tag{128}$$

Note that $\bar{\eta} \leq \frac{\|\boldsymbol{\eta}\|_2}{\sqrt{m}} \leq c_0\|\mathbf{x}\|_2$ (cf. (8)). Combining (119), (124), (127), and (128), and setting $\delta = \frac{c}{n}$, we obtain that when $m = \Omega(k\log(\tilde{L}nr))$, with probability $1 - e^{-\Omega(m)}$, it holds that

$$\left|\hat{\mathbf{x}}^T(\check{\mathbf{W}} - \mathbb{E}[\check{\mathbf{W}}])\hat{\mathbf{x}}\right| \leq c\|\mathbf{x}\|_2. \tag{129}$$

Combining (104), (118), and (129), we obtain that when $m = \Omega(k\log(\tilde{L}nr))$, with probability $1 - e^{-\Omega(m)}$, it holds that

$$\hat{\mathbf{x}}^T\mathbf{V}\hat{\mathbf{x}} \leq \left(c + \gamma' + \beta'(\hat{\mathbf{x}}^T\bar{\mathbf{x}})^2\right)\|\mathbf{x}\|_2. \tag{130}$$

**Combining and simplifying:** Finally, combining (99) and (130), we obtain that when $m = \Omega(k \log(\tilde{L}nr))$, with probability $1 - e^{-\Omega(m)}$,

$$(\gamma + \beta - c)\|\mathbf{x}\|_2 \leq \left(c + \gamma' + \beta'(\hat{\mathbf{x}}^T\bar{\mathbf{x}})^2\right)\|\mathbf{x}\|_2. \tag{131}$$

Simplifying (131), we obtain

$$\beta'(\hat{\mathbf{x}}^T\bar{\mathbf{x}})^2 \geq \beta + (\gamma - \gamma') - 2c. \tag{132}$$

Defining $\Phi(x) = \int_{-\infty}^{x} \frac{1}{\sqrt{2\pi}} e^{-\frac{t^2}{2}} \, dt$ to be the standard normal distribution function and setting $C_0 = \max_{x \in \mathbb{R}} \Phi'(x) = \frac{1}{\sqrt{2\pi}}$, we have for $g \sim \mathcal{N}(0,1)$ that

$$\gamma' - \gamma = \mathbb{E}[|g|\mathbf{1}_{\{l(1-c) \leq |g| \leq l(1+c)\}}] + \mathbb{E}[|g|\mathbf{1}_{\{u(1-c) \leq |g| \leq u(1+c)\}}] \tag{133}$$

$$\leq 2\mathbb{E}[|g|\mathbf{1}_{\{u(1-c) \leq |g| \leq u(1+c)\}}] \tag{134}$$

$$\leq 4u(1+c)\mathbb{P}(u(1-c) < g < u(1+c)) \tag{135}$$

$$\leq 5u(\Phi(u(1+c)) - \Phi(u(1-c))) \tag{136}$$

$$\leq 10C_0 u^2 c = O(c), \tag{137}$$

where (134) uses the definitions of $\gamma$ and $\gamma'$ following (89), (135) multiplies by two due the replacement of $|g|$ by $g$ in the probability, and (136) holds for small enough $c$. Similarly, we have $\beta \geq \beta' - O(c)$. Recall from (73) that the event $\mathcal{E}$ occurs with probability $1 - e^{-\Omega(m)}$. Therefore, from (132), we obtain that when $m = \Omega(k \log(\tilde{L}nr))$, with probability $1 - e^{-\Omega(m)}$, it holds that

$$(\hat{\mathbf{x}}^T\bar{\mathbf{x}})^2 \geq 1 - c'. \tag{138}$$

Without loss of generality, we assume that $\hat{\mathbf{x}}^T\bar{\mathbf{x}} > 0$ (otherwise, we can similarly derive an upper bound for $\|\hat{\mathbf{x}} + \bar{\mathbf{x}}\|_2$ instead of for $\|\hat{\mathbf{x}} - \bar{\mathbf{x}}\|_2$). Then, we have

$$\|\hat{\mathbf{x}} - \bar{\mathbf{x}}\|_2^2 = 2(1 - \hat{\mathbf{x}}^T\bar{\mathbf{x}}) \leq 2\left(1 - (\hat{\mathbf{x}}^T\bar{\mathbf{x}})^2\right) \leq 2c'. \tag{139}$$

By suitably renaming the constant, we obtain (16) as desired. Using Lemma 7, we also obtain (17).

## D  Proof of Lemma 6 (Variant of the S-REC)

For fixed $\delta > 0$ and a positive integer $\ell$, let $M = M_0 \subseteq M_1 \subseteq \ldots \subseteq M_\ell$ be a chain of nets of $B_2^k(r)$ such that $M_i$ is a $\frac{\delta_i}{L}$-net with $\delta_i = \frac{\delta}{2^i}$. There exists such a chain of nets with [60, Lemma 5.2]

$$\log|M_i| \leq k \log \frac{4Lr}{\delta_i}. \tag{140}$$

By the $L$-Lipschitz assumption on $G$, we have for any $i \in [\ell]$ that $G(M_i)$ is a $\delta_i$-net of $G(B_2^k(r))$. We write

$$\mathbf{x}_1 = (\mathbf{x}_1 - \mathbf{s}_\ell) + \sum_{i=1}^{\ell}(\mathbf{s}_i - \mathbf{s}_{i-1}) + \mathbf{s}_0, \quad \mathbf{x}_2 = (\mathbf{x}_2 - \mathbf{t}_\ell) + \sum_{i=1}^{\ell}(\mathbf{t}_i - \mathbf{t}_{i-1}) + \mathbf{t}_0, \tag{141}$$

where $\mathbf{s}_i, \mathbf{t}_i \in G(M_i)$ for all $i \in [\ell]$, and $\|\mathbf{x}_1 - \mathbf{s}_\ell\| \leq \frac{\delta}{2^\ell}$, $\|\mathbf{x}_2 - \mathbf{t}_\ell\| \leq \frac{\delta}{2^\ell}$, $\|\mathbf{s}_i - \mathbf{s}_{i-1}\|_2 \leq \frac{\delta}{2^{i-1}}$, and $\|\mathbf{t}_i - \mathbf{t}_{i-1}\|_2 \leq \frac{\delta}{2^{i-1}}$ for all $i \in [\ell]$. Therefore, the triangle inequality gives

$$\|\mathbf{x}_1 - \mathbf{s}_0\|_2 < 2\delta, \quad \|\mathbf{x}_2 - \mathbf{t}_0\|_2 < 2\delta. \tag{142}$$

For any index $I \subseteq [m]$ with $|I| \geq \frac{m}{2}$, from the triangle inequality, we have

$$\|\mathbf{A}_I(\mathbf{x}_1 - \mathbf{x}_2)\|_2 \geq \|\mathbf{A}_I(\mathbf{s}_0 - \mathbf{t}_0)\|_2 - \sum_{i=1}^{\ell}(\|\mathbf{A}_I(\mathbf{s}_i - \mathbf{s}_{i-1})\|_2 + \|\mathbf{A}_I(\mathbf{t}_i - \mathbf{t}_{i-1})\|_2)$$

$$\quad - \|\mathbf{A}_I(\mathbf{s}_\ell - \mathbf{x}_1)\|_2 - \|\mathbf{A}_I(\mathbf{t}_\ell - \mathbf{x}_2)\|_2 \tag{143}$$

$$\geq \|\mathbf{A}_I(\mathbf{s}_0 - \mathbf{t}_0)\|_2 - \sum_{i=1}^{\ell}(\|\mathbf{A}(\mathbf{s}_i - \mathbf{s}_{i-1})\|_2 + \|\mathbf{A}(\mathbf{t}_i - \mathbf{t}_{i-1})\|_2)$$

$$\quad - \|\mathbf{A}(\mathbf{s}_\ell - \mathbf{x}_1)\|_2 - \|\mathbf{A}(\mathbf{t}_\ell - \mathbf{x}_2)\|_2. \tag{144}$$

For any $\mathbf{r}_1, \mathbf{r}_2 \in G(M)$ and $0 < \alpha < \mu$, from Lemma 5, we have with probability $1 - e^{-\Omega(\alpha^2 m)}$ that

$$\|\mathbf{A}_I(\mathbf{r}_1 - \mathbf{r}_2)\|_2 \geq \sqrt{m}(\mu - \alpha)\|\mathbf{r}_1 - \mathbf{r}_2\|_2. \tag{145}$$

Taking a union bound over $G(M) \times G(M)$, we have that when $m = \Omega\left(\frac{k}{\alpha^2} \log \frac{Lr}{\delta}\right)$, with probability $1 - e^{-\Omega(\alpha^2 m)}$, for *all* $\mathbf{r}_1, \mathbf{r}_2 \in G(M)$,

$$\|\mathbf{A}_I(\mathbf{r}_1 - \mathbf{r}_2)\|_2 \geq \sqrt{m}(\mu - \alpha)\|\mathbf{r}_1 - \mathbf{r}_2\|_2. \tag{146}$$

Therefore, we have that when $m = \Omega\left(\frac{k}{\alpha^2} \log \frac{Lr}{\delta}\right)$, with probability $1 - e^{-\Omega(\alpha^2 m)}$,

$$\frac{1}{\sqrt{m}}\|\mathbf{A}_I(\mathbf{s}_0 - \mathbf{t}_0)\|_2 \geq (\mu - \alpha)\|\mathbf{s}_0 - \mathbf{t}_0\|_2 \geq (\mu - \alpha)(\|\mathbf{x}_1 - \mathbf{x}_2\|_2 - 4\delta), \tag{147}$$

where we applied (142). In addition, using the results in [5, 35], for the standard Gaussian matrix $\mathbf{A}$, we have $m = \Omega\left(\frac{k}{\alpha^2} \log \frac{Lr}{\delta}\right)$, with probability $1 - e^{-\Omega(\alpha^2 m)}$,

$$\sum_{i=1}^{\ell} (\|\mathbf{A}(\mathbf{s}_i - \mathbf{s}_{i-1})\|_2 + \|\mathbf{A}(\mathbf{t}_i - \mathbf{t}_{i-1})\|_2) \leq O(\sqrt{m}\alpha\delta) = O(\sqrt{m}\delta). \tag{148}$$

Moreover, by [60, Corollary 5.35], we have $\left\|\frac{1}{\sqrt{m}}\mathbf{A}\right\|_{2 \to 2} \leq 2 + \sqrt{\frac{n}{m}}$ with probability at least $1 - e^{-m/2}$. Hence, choosing $\ell = \lceil \log_2 n \rceil$, we have with probability at least $1 - e^{-m/2}$ that

$$\left\|\frac{1}{\sqrt{m}}\mathbf{A}(\mathbf{t}_\ell - \mathbf{x}_2)\right\|_2 + \left\|\frac{1}{\sqrt{m}}\mathbf{A}(\mathbf{s}_\ell - \mathbf{x}_1)\right\|_2 \leq 2\left(2 + \sqrt{\frac{n}{m}}\right)\frac{\delta}{2^l} = O(\delta), \tag{149}$$

where we used the fact that $\|\mathbf{t}_\ell - \mathbf{x}_2\| \leq \frac{\delta}{2^\ell}$ and $\|\mathbf{s}_\ell - \mathbf{x}_1\| \leq \frac{\delta}{2^\ell}$. Combining (144), (147), (148) and (149), we obtain the desired result.

# E   Further Experiments: Comparing the Iterative Algorithm in `CoPRAM` Combined with Various Initialization Methods

In this section, we provide additional numerical results comparing our approach with other initialization methods, in that case that the initialization is combined with the subsequent iterative algorithm used in `CoPRAM`. In addition to `PRI-SPCA-NT` and the spectral initialization methods used in `ThWF`, `SPARTA`, and `CoPRAM`, we also compare with random initialization, in which the initial vector is $\lambda \frac{\mathbf{g}}{\|\mathbf{g}\|_2}$ with $\mathbf{g} \sim \mathcal{N}(\mathbf{0}, \mathbf{I}_n)$. The worsened performance of random initialization reveals the importance of using spectral initialization methods for sparse phase retrieval.

We first consider the noiseless case and compare the relative error and empirical success rate of different methods. More specifically, after obtaining initial vectors from different initialization methods using the procedure discussed in Section 5, we run the iterative algorithm of `CoPRAM` for $T = 100$ iterations to further refine the estimated vectors and obtain $\mathbf{x}^{(T)}$. For our experiments, we found that 100 iterations are usually sufficient for convergence. A trial is declared to be successful if the relative error

$$\frac{\min\{\|\mathbf{x}^{(T)} - \mathbf{x}\|_2, \|\mathbf{x}^{(T)} + \mathbf{x}\|_2\}}{\|\mathbf{x}\|_2} \tag{150}$$

is less than 0.01. We fix $n = 1000$, and consider $s = 10$ or $s = 20$, while varying $m$ in $\{100, 150, 200, \ldots, 1000\}$. For each of the experiments in this section, we repeat 50 trials for 10 times, and calculate the standard deviation over these 10 times to produce the error bars.

We report the relative error in Figure 4, and we report the empirical success rate of different methods in Figure 5. From Figures 4 and 5, we observe that for most cases, our `PRI-SPCA` method leads to the smallest relative error and the largest empirical success rate.

Next, we consider the noisy case, and compare the relative error for using approximately the same time cost. The time cost is calculated as the sum of the running time of each initialization method and the running time of the subsequent iterative algorithm. As we have mentioned in Section 5, the time complexity of each iteration in `GRQI` is $O(s^3 + sn)$, while for the iterative algorithm of `CoPRAM`, the time complexity of each iteration is $O(s^2 n \log n)$, which dominates the total time complexity.

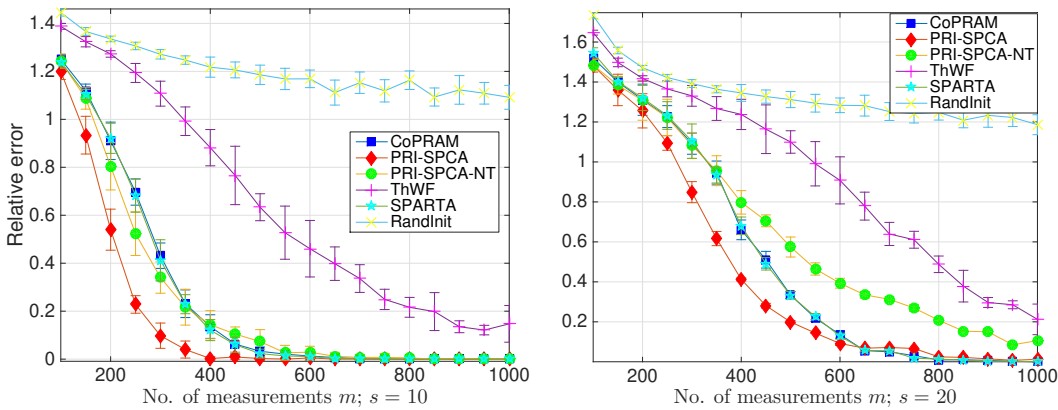

Figure 4: Relative error vs. number of measurements $m$ in the noiseless setting with $s = 10$ (Left) and $s = 20$ (Right).

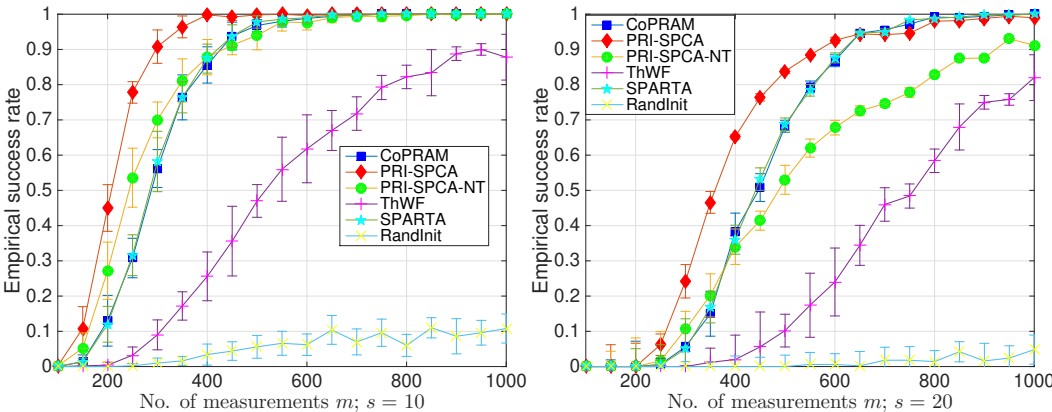

Figure 5: Empirical success rate vs. number of measurements $m$ in the noiseless setting with $s = 10$ (Left) and $s = 20$ (Right).

In this experiment, we fix $n = 1000$, $m = 500$, $s = 20$, and consider $\sigma = 0.1$ and $0.2$. Since the running times of the initialization methods are typically less than $0.1$ second, we vary the time cost (in seconds) $t$ in $\{0.1, 0.12, \ldots, 0.48, 0.5\}$. For each $t$ and each of the methods we consider, we find the number of the iteration whose time cost is closest to $t$, and record the corresponding relative error. Note that as mentioned in Section 5, for noisy measurements, we do not compare with `ThWF` because it corresponds to quadratic measurements. The results are reported in Figure 6, from which we observe that when combined with the iterative algorithm of `CoPRAM` and using approximately the same time cost, our `PRI-SPCA` method gives smallest relative error in most cases.

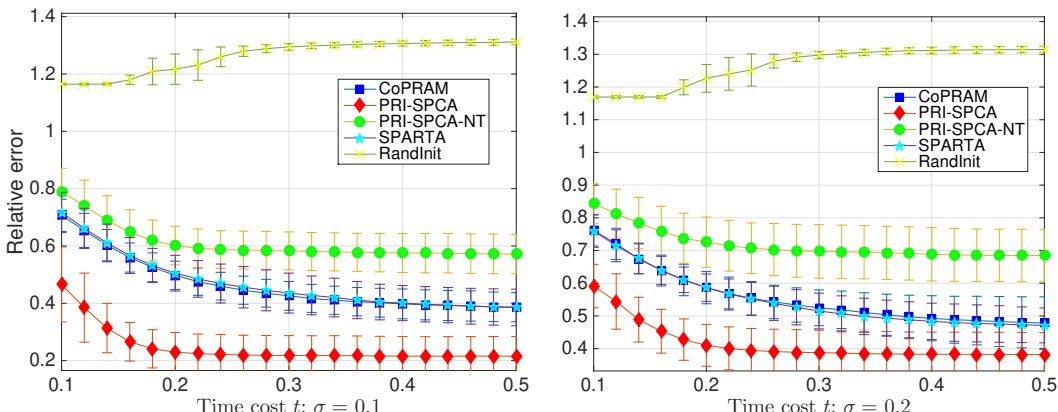

Figure 6: Relative error for different initialization methods combined with the iterative algorithm of CoPRAM in the noisy setting with $\sigma = 0.1$ (Left) and $\sigma = 0.2$ (Right), when using approximately the same time cost.