# OpenReview forum: "Towards Sample-Optimal Compressive Phase Retrieval with Sparse and Generative Priors"
_NeurIPS.cc/2021/Conference — NeurIPS 2021 Poster_

### Official Review · Reviewer_Uh34 · 2021-07-15

**Rating:** 6
**Confidence:** 4

**Summary:**

Summary:

This paper proposes a spectral initialization algorithm for compressive phase retrieval and show that it is provably close to the ground truth. They also generalize previous results in compressive phase retrieval using generative models. Finally, the authors show that sparse PCA can be used to find a good initialization for sparse phase retrieval.

**Ethical Concerns:**

I do not have concerns about the ethics of this paper as it is primarily theoretical.

**Limitations And Societal Impact:**

I do not have concerns about the social impact of this work as it is primarily theoretical.

**Main Review:**

__Strengths__:
1. All results and assumptions are stated clearly.

1. Prior work has analyzed the sample complexity of phase retrieval using $d$-layered ReLU networks. Theorem 1 shows that similar sample complexity is true for $L$-Lipschitz generative models. The result seems correct.

1. The authors propose a new spectral initialization algorithm, and their theoretical result shows that it is close to the true vector. This result can be combined with existing results in literature that depend on a good initialization to guarantee a polynomial time algorithm (under certain assumptions).

1. Algorithm 1 is interesting. Specifically, the use of sparse PCA to find an approximate initialization is interesting and novel to the best of my knowledge.

__Weaknesses__:

1. I think the claim that the authors have found an efficient algorithm for generative networks is overstated. The authors state that their spectral normalization result can be combined with the Theorem in [24] to get a polynomial time algorithm. However, the results in [24] assume that there exists a polynomial time algorithm to project onto the range of a generative model, and this is a strong assumption which is simply not true for many generative models. I think the authors should clarify this in their claims.

1. The novelty in the spectral initialization is unclear. Is it exactly the same as [66]?

1. [66] provides a sample complexity bound of $n \log n$ for phase retrieval of _general vectors_, and Theorem 3 in this paper provides a sample complexity bound of $s \log n$ for phase retrieval of _sparse vectors_. The authors should elaborate the novelty in Theorem 3. Can Theorem 3 be derived from [66] by replacing the metric entropy of general vectors with the metric entropy of sparse vectors?

1. The experiments are unconvincing. They are all synthetic experiments without a trained generative model, and it is unclear whether the proposed method using a generative model actually outperforms sparsity-based algorithms. Furthermore, most error bars overlap.

__Novelty__:

The spectral initialization used in this paper is derived from [66].

From my understanding of the paper, Theorem 1 follows from traditional analysis techniques in compressed sensing and phase retrieval. The case of $d$-layered ReLU generative networks have been explored in prior work, and this work generalizes it to $L$-Lipschitz generative models.

__Score Justification__:

I do have some concerns about the novelty of the results in this paper, as the spectral initialization technique and the other analysis techniques are largely borrowed from prior work. However, the authors do present a fairly complete analysis of the considered problem, and hence I think the merits outweigh the flaws.

__General comments__:
1. Below Eqn 9, the authors say that $\|\|\eta\|\|_\infty = O( \|\| \eta \|\|_2 / \sqrt{m})$ is a reasonable assumption. However, even in the simple case where $\eta$ is Gaussian, the correct bound would be $\|\|\eta\|\|_\infty = O(\|\| \eta \|\|_2  \sqrt{\log m }/ \sqrt{m})$.  I do not think it is a big problem since it is only an extra log-factor, but I think it should be stated correctly (and the Theorem statements should also account for this extra log factor, if necessary).

1. I understand that the assumption that range($G$) lies in the unit ball is necessary. However, if it lies within a $R$ radius ball, how do the bounds change? I think the dependence may be polynomial in $R$, but it would be nice to have an explicit formula.
Also, what does this assumption imply for the Lipschitz constant $L$? Does it need to be $1/r$?

1.

**Time Spent Reviewing:**

4

---

> ### Comment · Reviewer_Uh34 · 2021-08-18
> **Additional concern regarding prior work & experiments**
>
> ### Prior work
>
> In my original review, I was mainly comparing this paper with the prior work of [PH]. In [PH], the authors show that for a $d$-layered generative model $G:\mathbb{R}^k \to \mathbb{R}^n$, $O(kd^2 \log n)$ samples are enough for recovery; additionally, if the network has Gaussian random weights, then gradient descent on the input to $G$ with a random initialization will provably converge. In this submission, the authors are able to improve the bound slightly, as their complexity is $O(k \log L)$, and under the reasonable assumption that $L =  n^{O(d)}$, this means that their bound improves the bound in [PH] -- [PH] is quadratic in the depth $d$, while this submission is linear in $d$ (note that the dependence on $k$ is the same).
>
> However, I now recall that there is another prior work [BA]. The conclusion in [BA] is: "We compare the performance to sparse separable priors and conclude that in all cases analysed generative priors have a smaller statistical-to-algorithmic gap than sparse priors, giving theoretical support to previous experimental observations that generative priors might be advantageous in terms of algorithmic performance. In particular, while sparsity does not allow to perform compressive phase retrieval efficiently close to its information-theoretic limit, it is found that under the random generative prior compressed phase retrieval becomes tractable."
>
> There are some caveats: the generative model is random and the model is assumed to be in the high dimensional regime, i.e., $n,m,k\to \infty$, the ratio $m/n$ is kept fixed, and [BA] studies recovery feasibility as the ratio $k/n$ varies. However the main message of [BA] is that generative priors can perform recovery at the information-theoretic limit, and this is quite similar to that of this submission. [BA] and this submission also share the same weakness-- they cannot be run on real world networks.
>
> Overall I think this doesn't hurt the submission too much, since the algorithmic approach is different -- [BA] uses message passing, while this work seems to advocate spectral initialization for gradient descent. However, **this submission shouldn't claim to be the first to obtain optimal sample-complexity for generative phase retrieval**. This submission doesn't discuss the work of [BA], and I think it is important for the authors to compare their work with [BA].
>
> [PH] Hand, Paul, Oscar Leong, and Vladislav Voroninski. "Phase retrieval under a generative prior." In Proceedings of the 32nd International Conference on Neural Information Processing Systems, pp. 9154-9164. 2018.
>
> [BA] Aubin, Benjamin, Bruno Loureiro, Antoine Baker, Florent Krzakala, and Lenka Zdeborová. "Exact asymptotics for phase retrieval and compressed sensing with random generative priors." In Mathematical and Scientific Machine Learning, pp. 55-73. PMLR, 2020.
>
>
> ### Experiments
> I missed this in my original reading of the paper, but in Lines 259-261, the authors say "We focus only on the sparse setting, since numerous existing algorithms are known for sparse priors but not for generative priors.", and in their author response they say "We appreciate that experiments for generative priors are also of interest, but in this case, unlike for sparse priors, we are not aware of any related approaches (beyond random initialization) to compare against."
>
> Even if there were baselines, the authors would not be able to compare their algorithm, since the initialization for generative models is even more complicated than the sparse case (the sparse case is already computationally difficult since sparse PCA is inapproximable). I think there should be more discussion about the practical limitations of your work here, rather than citing the lack of baselines.

---

> > ### Author Response · Authors · 2021-08-19
> > **Responses to new comments**
> >
> > Thanks again for the comments. Regarding the prior work and experiments, our responses are as follows.
> >
> > (**Prior work**) This additional reference [BA] is very helpful, and we will definitely discuss it in the revised version. Indeed, both [PH] and [BA] obtain certain order-optimal sample complexity guarantees for generative phase retrieval, and we will avoid giving the reader the impression that our work is the first to do so. On the other hand, we believe that the settings of [BA] (and [PH]) are quite different from ours:
> >
> > (1) [PH] assumes a ReLU neural network generative model with no offsets, and all the weight matrices of the ReLU neural network are assumed to have i.i.d. zero-mean Gaussian entries. In addition, the neural network needs to be sufficiently expansive in the sense that $n_i \ge \Omega(n_{i-1}\log n_{i-1})$, where $n_i$ is the number of neurons in the $i$-th layer. [BA] maintains i.i.d. Gaussianity but slightly relaxes to general activation functions (not only ReLU) and $n_i \ge \Omega(n_{i-1})$.  Both works also focus on the noiseless case, though [BA] states that the noisy extension is possible.
> >
> > In contrast, in our work, we only use the much milder assumption that the generative model is Lipschitz continuous, with no assumptions on expansion properties, Gaussianity, etc.  Of course, this is not without some degree of trade-off (assumptions vs. results) -- [PH]’s assumptions additionally allow further deducing a favorable optimization landscape, which isn’t the case in our setting, and [BA]’s assumptions additionally allow a result with precise constants instead of just scaling laws.
> >
> > (2) As the reviewer suggests, the algorithmic approaches in [PH] and [BA] are very different from ours. We discuss in Section 1.1 that [PH] minimizes the objective function directly over the latent variable $\mathbf{z} \in \mathbb{R}^k$ using gradient descent. In our setting, the corresponding objective function is typically highly non-convex, which can cause issues with local minima in general optimization landscapes. The approximate message passing approach in [BA] is also of significant interest, but is very different from our study of spectral initialization for gradient descent over the ambient space in $\mathbb{R}^n$.
> >
> > (3) As the reviewer also hints, the authors of [BA] consider the high dimensional regime where $n,m,k \to \infty$ and the ratio $m/n$ is kept fixed, and they provide an asymptotic analysis (not yet established as fully rigorous). We provide a rigorous analysis with no restrictions on the ratio $m/n$. Moreover, in [BA], the latent variable is assumed to be drawn from a separable distribution $\mathbf{z}  \sim P_{\mathbf{z} }$, whereas in our work, the latent vector can be an arbitrary fixed vector in $B_2^k(r)$.
> >
> > Thanks again, and we will make these comparisons as clear as possible in the revised paper.
> >
> > (**Experiments**) This is also a helpful suggestion -- we will reduce the emphasis on the lack of baselines, and highlight that generative priors may pose further practical difficulties over the sparse setting.  On the other hand, it would be interesting to see if a gradient-based approach could also perform well under a generative prior even when the global optimum is not found.

---

### Official Review · Reviewer_b8Ro · 2021-07-15

**Rating:** 7
**Confidence:** 4

**Summary:**

This paper considers the problem of noisy compressive phase retrieval (CPR) of $n$ dimensional signal under two different priors: (i) $L$-Lipschitz continuous generative model with latent dimension $k$ and (ii) underlying sparsity $s$. The main claims of this paper are theoretical in nature, where the authors derive the sample requirements for existence of a solution for CPR under both types of priors (assuming there exists an algorithm that can  solve the CPR minimization). The sample requirements obtained are of the order required for a linear compressive sensing problem, and therefore roughly sample optimal.

Additionally the authors use a _truncated_ spectral initialization algorithm for CPR, which they show has solution that lies close enough to the ground truth signal using only O($k\log L$) (and O($s\log n$)) samples.

The authors design a related spectral initialization technique which can be combined with any other off-the-shelf CPR algorithms, while requiring fewer sample requirements then prior state of art.

Spectral initialization has been previously used for the general phase retrieval problem as well as under sparsity and structured sparsity priors, however incurring increased sample requirements. In this paper the authors prove closeness of a truncated spectral initialization to ground truth in sparse phase retrieval which brings down the sample requirement from O($s^2 \log n$) to O($s\log n$).

They also show similar guarantees for the case of deep generative priors.

**Limitations And Societal Impact:**

Limitations: No experiments on CPR using spectral initialization for signals with deep generative prior.

**Main Review:**

The main claims and contributions of this paper are as follows:

1. existence of a solution to the CPR problem using O($k \log L$) samples within small error margin. [Theorem 1]
2. spectral initialization for CPR with deep generative priors (only theoretical, no experiments)
3. sample optimality has been shown for local convergence of CPR with deep generative priors, however this paper introduces an alternate procedure for spectral initialization, which brings down the sample requirement for initialization and therefore overall global convergence of CPR. [Theorem 2]
4. improved sample complexity for spectral initialization in sparse phase retrieval. [Theorem 3]

[Theorem 1]
Authors provide guarantee for existence of a solution for CPR. If ground truth $x \approx p = G(z)$ and $q = G(\hat{z})$ that minimizes the phase retrieval objective up to additive term $\tau$ (assuming there exists some algorithm that can achieve this), then the authors provide theoretical guarantees to show that $distance(q,x)$ is upper bounded by $distance(p,x)$ up to an additive error factor.  Such guarantees have been shown for the problem of compressed sensing before [4], but is novel in the case of compressed phase retrieval.

[Theorem 3]
Spectral initialization for the case of sparse phase retrieval has been previously solved [27,58] by first indentifying the sparse support based on the diagonal entries of matrix $V$ from this paper and then performing PCA (or power method) on this reduced support to identify the top singular vector of V. The previous approach is approximate, and therefore if solved exactly can yield better sample complexity. This is the key contribution in this paper where they derive the sample requirement for the spectral initialization problem, by assuming that the one can tractably solve for the top singular vector of $V$ that lies in the signal prior set (s-sparse signals or k,L generative signals).

[Theorem 2]
The authors also provide theoretical conditions under which such spectral initialization produces a good initialization when deep generative priors are considered. Both Theorems 2 and 3 use covering arguments and lemmas from previous derivations which have been properly presented.

[SPCA for initialization]
Authors use off the shelf SPCA solvers such as GRQI and TPower methods to solve the spectral initialization problem.

The paper is very well written, contributions are clearly highlighted and literature review is excellent. The results and derivations for sparse phase retrieval are convincing.

The only set-back of this paper is that there are no experiments for CPR using deep generative priors (with spectral initialization). How would one go about designing a deep generative prior (DGP) constrained version of PCA? This paper will benefit from such discussion.

It has also been observed in several papers that random initialization of latent space vector ($z if x=G(z)$) (or network weights) suffices for solving CPR with deep generative (or untrained) networks for local and possibly global convergence. It would be interesting to see how spectral initialization with DGP version of PCA would fare against random initialization.

**Time Spent Reviewing:**

6 hours

---

### Official Review · Reviewer_yozY · 2021-07-22

**Rating:** 7
**Confidence:** 3

**Summary:**

This paper presents interesting theoretical analysis for compressive phase retrieval under sparse and generative priors for the signals. The results are sample optimal and suggest that O(k log L) iid Gaussian measurements are sufficient to guarantee near-perfect signal recovery, where L is the network Lipschitz constant.The paper also discusses a spectral initialization approach with generative priors.

**Limitations And Societal Impact:**

- Very little discussion on limitations.

**Main Review:**

+ The paper presents some new theoretical result for the recovery of a signal from amplitude measurements under generative prior assumption. The sample complexity seems similar to the earlier results in [24], but this paper uses different assumption/process for initialization.

+ The paper presents analysis of spectral initialization with generative prior-based constraints. This is an interesting contribution.

 - The paper also presents analysis for sparse phase retrieval and includes experiments and comparison with existing methods.

- The authors do not provide any algorithm to recover such initialization, nor do they provide any simulations for generative prior-based phase retrieval. It will be great if the authors can provide some additional insights and experiments for spectral initialization and recovery under generative priors.

- Overall, the paper is well written.

**Time Spent Reviewing:**

4

---

### Author Response · Authors · 2021-08-09
**Thanks for the three anonymous reviewers**

We are very grateful to the reviewers for their helpful feedback and suggestions, and are pleased to have received a positive response. Our responses to the main concerns are given as follows.


[**Response to all reviewers**]

(**Experiments for generative priors**)  We appreciate that experiments for generative priors are also of interest, but in this case, unlike for sparse priors, we are not aware of any related approaches (beyond random initialization) to compare against.  In addition, our work is primarily theoretical, with the experiments serving as a proof of concept rather than seeking to be comprehensive.  We believe that the current experiments are sufficient for this purpose, and hope that our findings can motivate further practical/experimental work in the future.


[**Response to Reviewer Uh34**]

(**Overstated claim**) We wish to be open about the fact that both our spectral initialization approach and the projection onto the range of a generative model in [24] require some level of approximation (or strong assumptions such as exact projection).  We will accordingly modify the relevant discussions, toning down statements and clarifying the combining of these two approaches. We additionally searched our paper for the words  “efficient” and “polynomial time” and did not notice other overstated claims, but if the reviewer lets us know any relevant locations, we will further edit to avoid anything misleading.

(**The novelty compared to [66]**) Our work and [66] both use the truncated empirical matrix $\mathbf{V}$ in Eq. (11), but proceed in different ways. We highlight the following:

(i) We consider a constrained optimization problem instead of unconstrained, e.g., reducing to SPCA in the sparse case, and to our knowledge, no SPCA-like approaches to initialization with order-optimal sample complexity have been considered previously.

(ii) The most relevant result in [66] is Proposition 1, as that proposition concerns the initialization part. However, its proof is much simpler than ours ($2$ pages in their Appendix B), and does not include covering numbers (which are only mentioned in a less related part).  We do not see a way to insert covering numbers or metric entropy into its proof.

(iii) We use a covering argument with distance $\delta = O(1/n)$, but doing this for general signals would lead to $O(n \log n)$ sample complexity, instead of $O(n)$ in [66]. This further suggests that the approaches are fundamentally different.  (Note: The extra log factor is not a weakness in our setting, because it is known to be unavoidable, e.g., see [34])

To mention a possible alternative approach, to apply results for general phase retrieval to sparse phase retrieval, a common approach is to first estimate the support of the underlying sparse signal, and then reducing the sparse phase retrieval problem to a general phase retrieval problem within this estimated support. However, as we can see from some existing literature such as [6,27,39], the sample complexity of an accurate estimate of the support is typically at least $O(s^2 \log n),$ instead of $O(s \log n)$.

(**Error bars**) The error bars indeed overlap in some plots (less so in the appendix plots), but even in such cases, we believe there is a clear ordering of the performance.  As a toy example, if the “relative error random variable” changes from “uniform between $0.15$ to $0.25$” to “uniform between $0.1$ to $0.2$”, then the error bars will overlap, but the latter is clearly preferable.

(**The assumption in Eq. (9)**) Since the proof of Theorem 2 is already quite lengthy, we believe that (9) provides the best trade-off of generality and analytical tractability/brevity; it is similarly adopted in earlier works such as [9,65].  The Gaussian case indeed doesn’t satisfy the example $\\|\mathbf{\eta}\\|_\infty = O(\\|\mathbf{\eta}\\|_2/\sqrt{m})$ we provided for (9), but any “truncated Gaussian” does.

To circumvent this issue for the case that $\mathbf{\eta}$ is Gaussian with $\mathbf{\eta} \sim \mathcal{N}(\mathbf{0},\sigma^2 \mathbf{I}_m)$, the simplest approach would be to assume that $\sigma \sqrt{\log m} \le c_0 \\|\mathbf{x}\\|_2$ to replace Eq. (8) (and remove Eq. (9)). The rest of the analysis remains identical. We will add such discussion concerning random noise in the revised version, and highlight that further avoiding the extra $\sqrt{\log m}$ factor in this assumption would be ideal but is beyond our scope.

(**The assumption on $\mathrm{Range}(G)$**) For our Theorem 2, the range of the generative model $G$ can be a large set in $\mathbb{R}^n$, since we will always consider its normalized version $\tilde{G}$, whose range is contained in the unit sphere in $\mathbb{R}^n$ (see the paragraph below Eq. (10)). Also, $r$ can be of the same order as $L$, which is typically of scaling $n^{O(d)}$ for a $d$-layer neural network.


[**Response to Reviewer yozY**]

(**Limitations**) We have made an effort to make the limitations clear, including the fact that a complete solution for an efficient algorithm still remains open.  We will also further revise to make sure they are highlighted sufficiently, including the one outlined above regarding [24], and the fact that our experiments serve primarily as a proof of concept rather than a comprehensive empirical study.

---

### Decision · Program_Chairs · 2021-09-27

**Decision:**

Accept (Poster)

**Comment:**

This paper studies the problem of phase retrieval with generative priors. First they show under Gaussian measurements and a generative model that is $L$-Lipschitz and has $k$ inputs, $O(k \log L)$ measurements suffice for recovery. Furthermore they give an algorithm, predicated on solving a particular spectral initialization problem, that actually attains this number of measurements. There is a catch here: The optimization problem itself is likely hard. This is not surprising. After all, even in sparse phase retrieval there is conjectured to be a computational vs. statistical gap (and evidence is known based on the planted clique problem). Compared to the closely related work of Hand-Leong-Voroninski the main differences are (1) Hand et al. work with $d$-layer networks and get suboptimal dependence on $d$, whereas the bounds here are optimal up to constant factors (2) Hand et al. require an expansion condition on the network, which is not needed here. However (3) Hand et al. show that expansion gives a favorable optimization landscape, and thus have no hidden assumptions about being able to solve computationally hard problems.